# The host exosome pathway underpins biogenesis of the human cytomegalovirus virion

Declan L Turner[1], Denis V Korneev[2], John G Purdy[3], Alex de Marco[4,5,6], Rommel A Mathias[1,4]*

[1]Infection and Immunity Program, Monash Biomedicine Discovery Institute, Department of Microbiology, Monash University, Victoria, Australia; [2]School of Biological Sciences, Monash University, Victoria, Australia; [3]Department of Immunobiology and BIO5 Institute, University of Arizona, Tucson, United States; [4]Infection and Immunity Program, Monash Biomedicine Discovery Institute, Department of Biochemistry and Molecular Biology, Monash University, Victoria, Australia; [5]ARC Centre of Excellence in Advanced Molecular Imaging, Monash University, Victoria, Australia; [6]University of Warwick, Coventry, United Kingdom

**Abstract** Human Cytomegalovirus (HCMV) infects over half the world's population, is a leading cause of congenital birth defects, and poses serious risks for immuno-compromised individuals. To expand the molecular knowledge governing virion maturation, we analysed HCMV virions using proteomics, and identified a significant proportion of host exosome constituents. To validate this acquisition, we characterized exosomes released from uninfected cells, and demonstrated that over 99% of the protein cargo was subsequently incorporated into HCMV virions during infection. This suggested a common membrane origin, and utilization of host exosome machinery for virion assembly and egress. Thus, we selected a panel of exosome proteins for knock down, and confirmed that loss of 7/9 caused significantly less HCMV production. Saliently, we report that VAMP3 is essential for viral trafficking and release of infectious progeny, in various HCMV strains and cell types. Therefore, we establish that the host exosome pathway is intrinsic for HCMV maturation, and reveal new host regulators involved in viral trafficking, virion envelopment, and release. Our findings underpin future investigation of host exosome proteins as important modulators of HCMV replication with antiviral potential.

*For correspondence:
rommel.mathias@monash.edu

Competing interests: The authors declare that no competing interests exist.

## Introduction

Human Cytomegalovirus (HCMV) is a ubiquitous human pathogen that infects over half the world's population, with increased prevalence (>70%) in low socioeconomic demographics (*Cannon et al., 2010*). HCMV generally establishes an asymptomatic latent infection in healthy individuals, but poses a serious risk for those immuno-compromised due to organ transplantation or HIV infection (*Azevedo et al., 2015*; *Gianella and Letendre, 2016*). Additionally, congenital HCMV infection is a leading cause of birth defects, causing hearing, vision and cognitive impairments (*Manicklal et al., 2013*). With no vaccine currently available, and resistant strains emerging to current therapeutics (*Lurain and Chou, 2010*), the discovery of new antiviral targets is urgently required.

HCMV is a betaherpesvirus with a 236 kb double-stranded DNA genome encoding >170 open reading frames expressed during the viral replication cycle (*Murphy and Shenk, 2008*). The genome is housed within an icosahedral nucleocapsid surrounded by a dense layer of tegument protein (*Yu et al., 2017*), and an outermost lipid bi-layer forms the virion membrane. HCMV induces an elegant cascade of viral gene expression (over a 96–120 hr period in cell culture) to first replicate and

package the genome into nucleocapsids in the nucleus, and then navigate maturing nucleocapsids into the cytoplasm for tegumentation and envelope acquisition (*Gibson, 2008*). Whilst knowledge governing the early stages of the replication cycle and events that occur in the nucleus are known, considerably less can be said about cytoplasmic virion assembly and egress. A continuum of molecular processes is thought to facilitate this maturation, including formation of the viral assembly compartment (vAC), viral cargo recruitment, secondary envelopment, cytoplasmic trafficking, membrane fusion and release of infectious progeny (*Van Damme and Van Loock, 2014*; *Jean Beltran and Cristea, 2014*). However, the molecular players and mechanisms regulating this important stage of viral replication remain elusive.

A characteristic hallmark of HCMV-infected cells is the cytoplasmic megastructure known as the vAC (*Das and Pellett, 2011*; *Das et al., 2007*; *Sanchez et al., 2000a*). HCMV induces extensive host organelle remodelling (*Jean Beltran et al., 2016*), and around 72 hr post infection (HPI), infected cells display an enlarged kidney-bean-shaped nucleus, juxtaposed to a Golgi-derived vesicle ring that encapsulates endosome-positive vesicles (*Das and Pellett, 2011*; *Das et al., 2007*; *Sanchez et al., 2000a*). Several viral proteins including UL48, UL94 and UL103 have been shown to be involved in vAC generation (*Das et al., 2014*; *Ahlqvist and Mocarski, 2011*; *Phillips and Bresnahan, 2012*). Ultrastructural electron microscopy-based imaging demonstrated nucleocapsids budding into short cisternae or multivesicular bodies (MVBs), indicating the vAC provides the cellular architecture for secondary envelopment (*Schauflinger et al., 2013*; *Fraile-Ramos et al., 2002*). The tegument protein UL99 localizes in the vAC during infection and is essential for virion production (*Sanchez et al., 2000b*; *Silva et al., 2003*; *Seo and Britt, 2006*), and UL71 is important for secondary envelopment (*Womack and Shenk, 2010*; *Schauflinger et al., 2011*; *Dietz et al., 2018*). Additionally, several host proteins including CD63, TGN46, CI-M6PR/IGF2R and TFR have been reported as vAC residents during infection (*Cepeda et al., 2010*). As less is known about their precise functional involvement, host proteins have been the focus of recent investigations (*McCormick et al., 2018*).

Rather than generating entirely new cellular processes, HCMV beneficially hijacks existing host pathways (*Alwine, 2012*). Most cell types continuously secrete a variety of extracellular vesicles (EVs), including 100–1000 nm microvesicles (MVs), and 30–200 nm exosomes, which are critical mediators of cell communication and signalling (*Colombo et al., 2014*; *Raposo and Stoorvogel, 2013*; *Hessvik and Llorente, 2018*; *Kalluri and LeBleu, 2020*). Whilst MVs bud outwards from the plasma membrane, exosome biogenesis navigates a different route. Intraluminal vesicles (ILVs) bud inward from the limiting MVB membrane, and trafficking of the MVB to the plasma membrane, and subsequent fusion, releases the ILVs as exosomes into the extracellular space. The similarity of exosome biogenesis with viral assembly and egress prefaces the finding that many herpesviruses hijack the exosome pathway for various aspects of viral pathogenesis (*Sadeghipour and Mathias, 2017*). Host Rab GTPases that regulate cellular traffic and fusion, have been implicated in HCMV replication as disruption of RAB4B, RAB11, and RAB27A significantly limited virus production (*McCormick et al., 2018*; *Krzyzaniak et al., 2009*; *Fraile-Ramos et al., 2010*). The endosomal sorting complex required for transport (ESCRT) pathway comprising five distinct complexes (ESCRT-0, -I, -II, -III and the VPS4 complex) (*Schmidt and Teis, 2012*) is the prototypical mechanism known for ILV formation, mediating cargo selection, and membrane budding and scission (*Christ et al., 2017*). However, the involvement of ESCRTs during HCMV replication remains unclear. Whilst TSG101 and ALIX/PDCD6IP have been shown to be dispensable for HCMV production, the ESCRT-III complex and VPS4 phenotypes appear to be somewhat dependent upon the inhibition system employed (*Fraile-Ramos et al., 2007*; *Tandon et al., 2009*; *Streck et al., 2018*). Nonetheless, the emergence of ESCRT-independent ILV formation mechanisms including tetraspanins, ceramide/sphingomyelinase, and syndecans (*Hessvik and Llorente, 2018*), hint towards the possibility that other host machinery may provide a more significant and robust contribution to HCMV virion maturation.

In this study we used a reverse-engineering strategy to first define the host protein components in the HCMV virion, and then reconcile their involvement in virion maturation. A significant enrichment of host exosome proteins was evident, however, these were not the classical exosome markers and ESCRT machinery previously investigated. Instead, our unbiased screening approach indicated that HCMV had co-opted the host exosome pathway using a novel nexus of molecular players, including VAMP3. Furthermore, in addition to releasing infectious virions, infected cells also released exosomes with specific infection-induced molecular cargo. Therefore, our findings cast the host

exosome biogenesis pathway as a crucial pathogenic requirement for viral cellular processes that underpin HCMV virion assembly and egress.

## Results

### HCMV virions are enriched with host exosome proteins

HCMV requires the host cellular architecture to assemble and release infectious virions. We therefore reasoned that analyzing host proteins within purified virions may reveal insights into their biogenesis and egress.

First, we collected conditioned medium and infected cells. Cells were sonicated, and the supernatant released was pooled with the conditioned medium for separation on a glycerol-tartrate gradient. Virions, Non-Infectious Enveloped Particles (NIEPs), and Dense Bodies (DBs) resolved clearly, and each band was individually extracted and further enriched on a second gradient (*Figure 1—figure supplement 1A*). Next, to minimize cytosolic contamination, we omitted the sonicated cell supernatant as input, and separated conditioned medium only. Whilst virions and NIEP bands remained prominent on the gradient, significantly less DBs were present (*Figure 1—figure supplement 1B*). To quantify this result further, we collected conditioned medium only, and imaged virions, NIEPs and DBs by Cryo-EM (*Figure 1—figure supplement 1C*). Quantitation of each particle type identified a total of 274 virions, 104 NIEPs and 44 DBs across 20 fields of view (*Figure 1—figure supplement 1D*). Conditioned medium only was used as input for all subsequent experiments. Proteomic analysis of virions extracted from the second gradient identified 69 viral and 2704 host proteins (*Figure 1A*, *Supplementary file 1*). The number of viral proteins identified was similar to a previous proteomic analysis of the HCMV virion in 2004 that reported 71 viral and 70 host proteins (*Varnum et al., 2004*). However, the number of host proteins was considerably higher (*Figure 1A*). Quantitative analysis revealed that viral protein abundance was 36%, whilst host proteins were dominant with 64% (*Figure 1B*).

To further confirm that cellular debris from lysed/dead cells did not contaminate conditioned medium-only virion preparations, we performed proteomics on infected cellular lysate (ICL) and plotted the relative abundance of each protein against its relative virion abundance (*Figure 1C*). As expected, most structural virion components including the major capsid protein (MCP), UL32 and glycoprotein B (gB) were more enriched in the virion compared to the ICL, while the viral polymerase processivity factor UL44 was enriched approximately 30-fold in the ICL. This analysis also revealed that individual host endosome and exosome proteins were more enriched in the virion, whilst some mitochondrial, nuclear, and endoplasmic reticulum marker proteins were enriched in the ICL (*Figure 1C*). To expand these observations to our global dataset, we calculated a normalized abundance for various host organelles (associated with HCMV assembly and egress) in ICL and virion samples, and computed a relative enrichment ratio. A ratio greater than one indicated enrichment in the virion compared to ICL, and if all host proteins identified in the virion were from lysed cell contamination, then all organelle ratios would be 1. However, exosome and endosome proteins were more enriched in virions, while proteins from the nucleus, ER and mitochondria were under-represented compared to their abundance in the ICL (*Figure 1D*). Additional gene ontology (GO) term analysis using DAVID software similarly confirmed the enrichment of many host proteins within virions as having 'extracellular exosome' cellular component annotation (*Figure 1E*).

To further validate that the presence of host exosome proteins in virions was not from cellular contamination or non-specific binding, we treated isolated virions with gentle surface shaving using proteinase K. This removed co-isolating proteins not incorporated within the virion membrane, as well as the outer-virion-facing domains of transmembrane/envelope proteins. Surface-shaved virions were then re-isolated on gradients to confirm virion integrity, and sequenced by mass spectrometry. Proteinase K treatment depleted virion proteins to 44 viral and 638 host components (*Figure 1F*, *Supplementary file 1*), and increased viral protein abundance to 82% (*Figure 1G*). The shift in balance was largely due to removal of high abundance cellular (actin, tubulin) and extracellular (collagens) host proteins but also included low abundance host proteins which were reduced below the detection threshold, evidenced by the lower total number detected. Shaving reduced the correlation coefficient to 0.363 (*Figure 1H*). This was caused by fewer host proteins detected, and greater relative abundance of viral proteins in the virion, compared to the ICL (*Figure 1H*). Strikingly, shaving

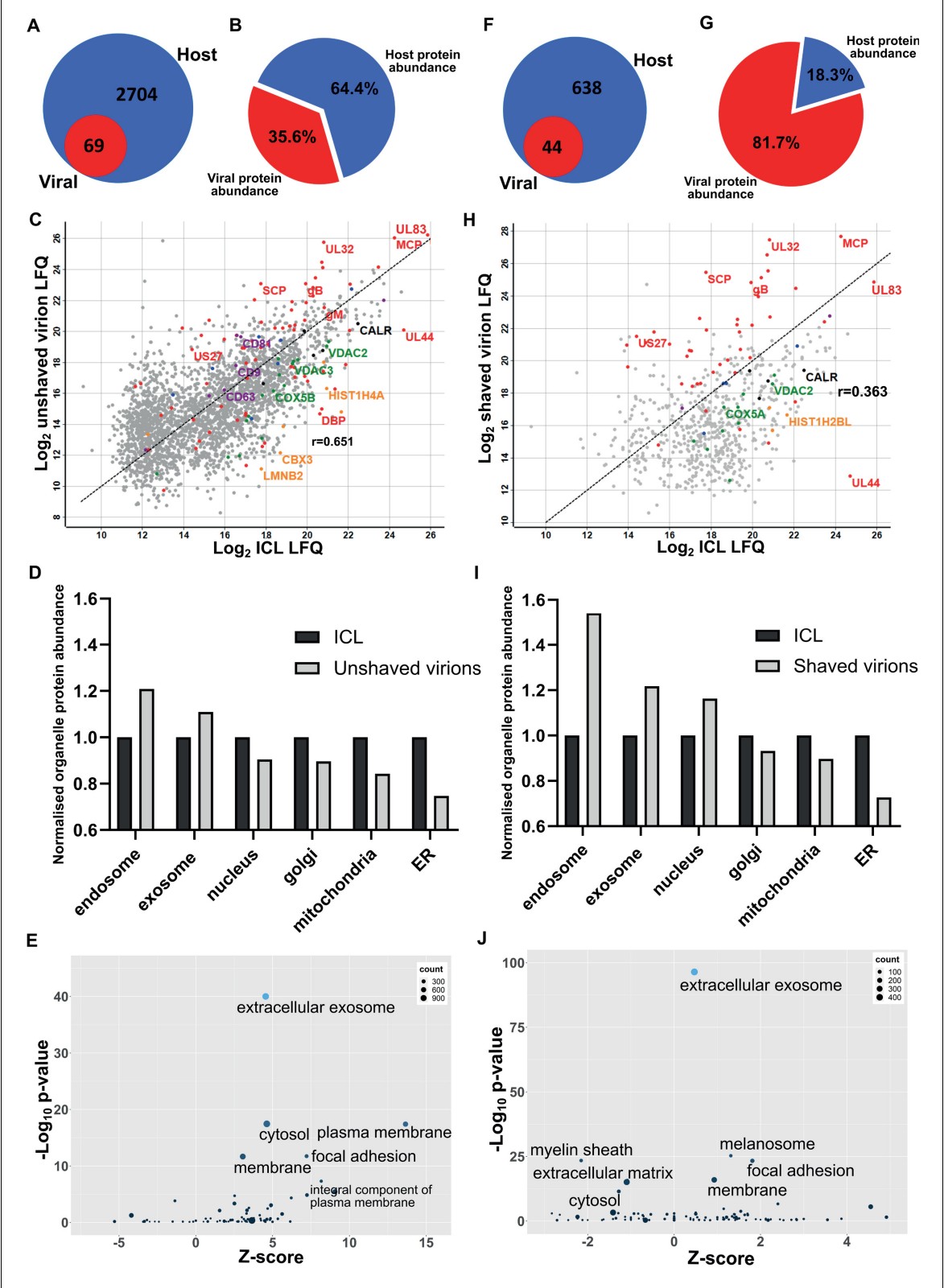

**Figure 1.** Proteomic analysis reveals HCMV virions are significantly enriched with host exosome proteins. (A) Total number of host and viral proteins identified in the virion by mass spectrometry. (B) Abundance proportions of host and viral proteins in the virion, as determined by MaxQuant label-free quantitation (LFQ). (C) Scatter plot of unshaved virion proteins versus HCMV infected cellular lysate (ICL), 5 DPI, MOI = 3. Red: HCMV proteins, green: select mitochondrial marker proteins, orange: nuclear markers, black: endoplasmic reticulum markers, blue: endosome markers, purple: exosome

*Figure 1 continued on next page*

Figure 1 continued

markers. Equation of line: y = x, r = 0.651. (D) Relative abundance of various host organelles in ICL compared to virions purified from infected cell culture supernatant. Total organelle abundance was calculated by summing individual protein abundance. (E) Gene ontology (cellular component) enrichment analysis of unshaved virions compared to ICL background. Analysis was performed using DAVID Functional Annotation and results plotted with GOPlot. (F) Number of host and viral proteins identified in the virion following proteinase K surface shaving. (G) Abundance proportions of host and viral proteins in surface shaved virions. (H) Scatter plot of all proteinase K shaved virion proteins versus HCMV ICL. Red: HCMV proteins, green: select mitochondrial marker proteins, orange: nuclear markers, black: endoplasmic reticulum markers, blue: endosome markers, purple: exosome markers. Equation of line: y = x, r = 0.363. (I) Relative abundance of various host organelles in ICL compared to proteinase K-shaved HCMV virions. (J) Gene ontology (cellular component) enrichment analysis of proteinase K shaved virions compared to ICL background. Performed with DAVID as for (E).

The online version of this article includes the following figure supplement(s) for figure 1:

**Figure supplement 1.** Separation of virions, NIEPs and Dense Bodies using glycerol-tartrate gradient centrifugation.

did not profoundly alter the cellular component distribution, but instead, increased the magnitude of the normalized enrichment ratio for endosome, exosome, and nuclear proteins (*Figure 1I*).The latter is expected as these proteins are associated with the viral genome inside the nucleocapsid or closely associated with the viral envelope, and therefore resistant to shaving. GO cellular component analysis with the shaved virion dataset revealed that 'cytosol' and 'plasma membrane' terms were less enriched and replaced by 'melanosome' (*Figure 1J*). 'Membrane', 'focal adhesion' and 'extracellular exosome' retained high confidence post shaving.

Collectively, this set of experiments identified a significant enrichment of host endosome and exosome proteins in mature HCMV virions. Their presence was not due to contamination or non-specific binding, as verified by enrichment ratio experiments as well as Proteinase K shaving experiments. Given exosomes are membranous nanovesicles secreted by almost all cells (*Colombo et al., 2014*), we hypothesized that HCMV may exploit this host pathway for virion maturation.

## Buoyant density can be used to separate exosomes and HCMV virions

To explore the intersection of host exosomes and virions experimentally, we established conditions to purify all vesicles released from fibroblasts with and without HCMV infection. Supernatants were collected from uninfected and infected cells, and separated across 10 density-increasing Optiprep gradient fractions (*Figure 2—figure supplement 1A–C*). Analysis of canonical exosome markers including ESCRTs and tetraspanins across uninfected and infected gradients revealed exosome markers were enriched in less-dense vesicle fractions (2 and 3). This is consistent with the expected density range for exosomes of 1.07–1.10 g/ml (*Xu et al., 2015*; *Figure 2—figure supplement 1D–E*). By contrast, viral proteins were enriched in more-dense fractions (7 and 8) on infected gradients (*Figure 2—figure supplement 1F*). Given our vesicles of interest were sufficiently separated, all 10 fractions from both uninfected and infected gradients were further analyzed by quantitative proteomics in biological triplicate. A total of 2012 proteins were identified (*Supplementary file 2*), and the relative abundance of each protein in each fraction of the gradient was calculated, for comparison across infected and uninfected conditions (*Figure 2—figure supplement 2*). Hierarchical clustering was also performed on all fractions using Pearson coefficients to visualise and confirm a robust fractionation process (*Figure 2—figure supplement 3*).

To more comprehensively evaluate the gradient fraction that exosomes resolved within, we compiled a list of 50 known exosome-associated markers, and used heatmaps to visualise their enrichment across the uninfected gradient. Consistent with western blotting analysis (*Figure 2—figure supplement 1D*), the majority of exosome marker proteins were primarily enriched in vesicles that resolved in fractions 1 and 2, and a smaller subset were also detected in fraction 7 (*Figure 2A*). The latter was not totally unexpected since the density of fraction 7 is 1.14–1.19 g/ml (*Figure 2—figure supplement 1A*), and some of the exosome protein markers are also known to be present in lower abundance in larger, denser MVs derived from the plasma membrane (e.g. CD63) (*Xu et al., 2015*). Therefore, given the lower relative enrichment of exosome markers, but high relative abundance of ECM proteins, it is most likely that MVs resolved in fraction 7 of the uninfected gradient.

Next, we characterized vesicle populations released from infected cells. Similar to the uninfected gradient samples, exosome markers were enriched in fractions 1 and 2 (*Figure 2—figure supplement 1E*). However, the markers were also strongly enriched in fraction 3, indicating that infection may have caused an increase to exosome density. Additionally, exosome markers were also more

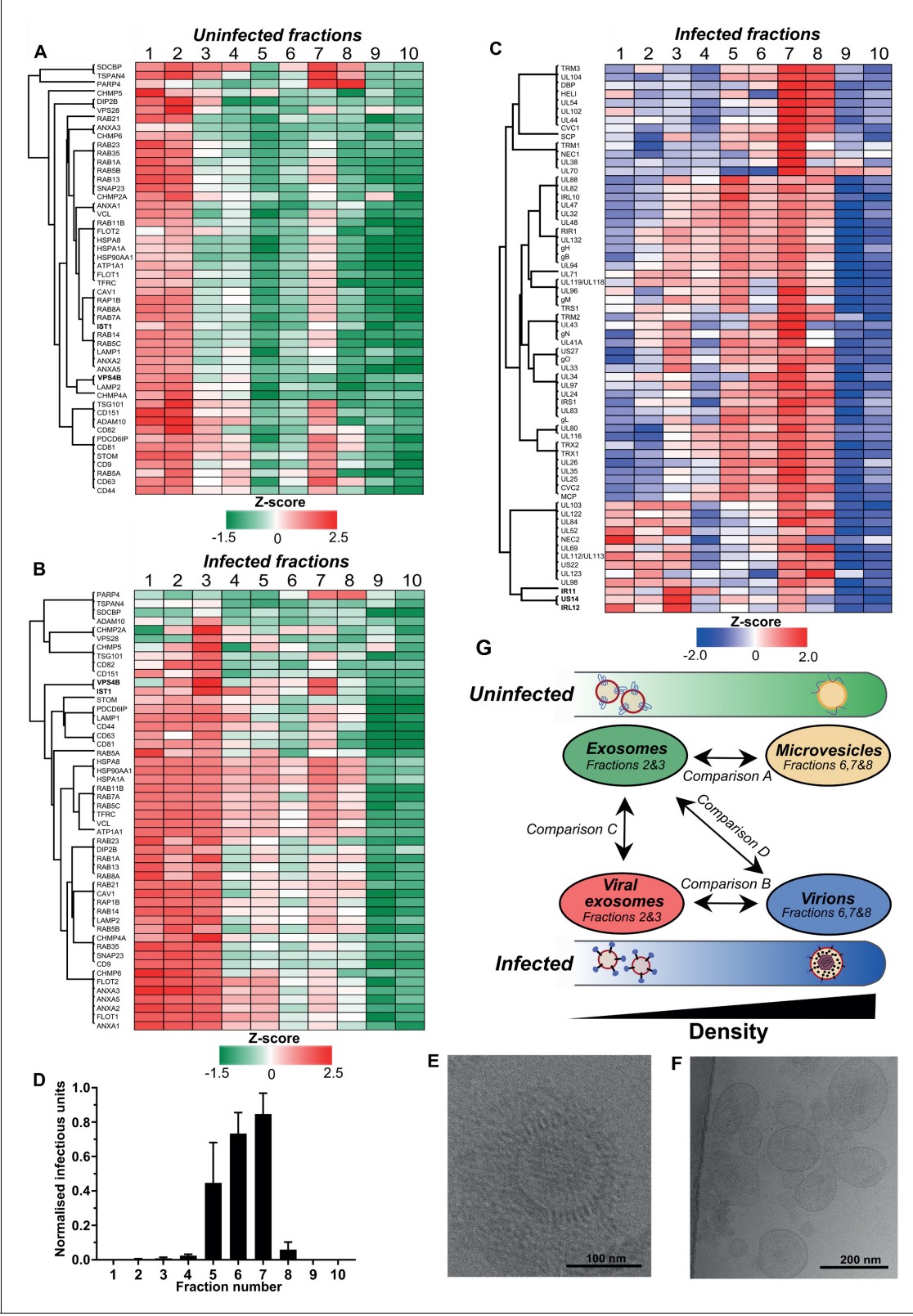

**Figure 2.** Exosomes and HCMV virions can be separated by their buoyant density. Supernatant from uninfected and infected cells was collected, resolved by Optiprep density gradient ultracentrifugation, 10 fractions (increasing density) extracted and proteins identified by mass spectrometry (n = 3 for each condition). Protein abundance was determined by MaxQuant label-free quantitation, and mean values from the triplicate were Z-scored across gradient fractions, and plotted using Perseus. Heatmap distribution of 50 exosome markers, across (**A**) uninfected gradient fractions, or (**B**)
*Figure 2 continued on next page*

*Figure 2 continued*

HCMV infected gradient fractions. (**C**) Enrichment of HCMV proteins across infected gradient fractions. (**D**) Infectious units in each fraction, as determined by the IE1 reporter plate assay (n = 4, bars = SEM). (**E-F**) Representative cryogenic transmission electron micrographs of infected gradient fractions 7 (E) and 3 (F). (**G**) Schematic summary of all vesicle populations that resolved across density gradients in the current study.

The online version of this article includes the following figure supplement(s) for figure 2:

**Figure supplement 1.** Separation and characterization of vesicles and particles resolved by OptiPrep gradients.
**Figure supplement 2.** Cluster analysis of all proteins detected from OptiPrep gradient fractions.
**Figure supplement 3.** Pearson correlation of all OptiPrep gradient fractions.
**Figure supplement 4.** Enrichment of selected proteins across density gradients and exosome refinement.

readily detected in more-dense fractions (4 to 8), and infection altered the expression distribution profile for some markers. For example, VPS4B and IST1 were mainly enriched in fraction two compared to fraction seven on the uninfected gradient (*Figure 2A*, *Figure 2—figure supplement 4A– B*). However, the enrichment of both proteins increased in fraction 7 of the infected gradient (*Figure 2B*, *Figure 2—figure supplement 4A–B*). This altered distribution caused by HCMV reflected a clear protein re-localization, and suggested that a subset of host exosome proteins were incorporated into a denser infection-specific entity.

To determine its identity, we evaluated expression of the 64 viral proteins identified by proteomics. Strikingly, all 64 viral proteins were detected in fraction 7, and most were predominantly enriched in this fraction, compared to less-dense fractions (*Figure 2C*). This indicated that infected fraction seven most likely contained virions. To validate this, we assayed the infectivity of each fraction across the gradient by incubating a small sample with uninfected cells in a reporter plate, and counted cells that expressed HCMV protein IE1 at 24 HPI. In agreement with the proteomics that determined most viral proteins were enriched in fraction 7, this fraction contained the most infectious units, whilst fractions 6 and 5 were also positive, but to a lesser extent (*Figure 2D*). Furthermore, examination of infected fraction seven by EM confirmed HCMV virions (*Figure 2E*).

The infectivity observed across neighboring fractions (*Lurain and Chou, 2010*; *Murphy and Shenk, 2008*; *Yu et al., 2017*; *Gibson, 2008*) could be due to varying virion density, or tailing effects resulting from gradient separation. However, virtually no infectious units were present in fractions 2 and 3 (*Figure 2D*), and EM analysis of infected fraction three also confirmed it was devoid of virions (*Figure 2F*). Instead, a heterogenous population of 60–200 nm membranous vesicles was observed (*Figure 2F*), which is the expected size for exosomes (*Xu et al., 2015*), and the heterogeneity in size is also consistent with a more diverse density spread across fractions 1–3 (*Figure 2B*). Therefore, we concluded that the viral proteins identified in fraction three were likely to be related to infected cell exosomes, and not from contaminating virions. As such, a small subset of viral proteins including IRL12, US14, and IR11/gp34 were enriched in fraction 3, compared to fraction 7 (*Figure 2C*, *Figure 2—figure supplement 4C*).

There was a small amount of protein present in infected fraction 10, but absent from the corresponding uninfected fraction (*Figure 2—figure supplement 1B–C*). Although viral proteins were identified by proteomics (*Figure 2C*, *Supplementary file 2*) and confirmed by western blotting (*Figure 2—figure supplement 1F*), no infectivity was observed (*Figure 2D*). We reasoned that DBs were likely to resolve in fraction 10 given the high density and low amount freely released into cell culture supernatant compared to virions (*Figure 1—figure supplement 1B–E*). To confirm this, the 15 most abundant viral proteins representing 96% of the DB proteome previously reported by *Varnum et al., 2004* was analyzed. In infected fraction 10, the 15 proteins comprise 83% of all viral protein in the fraction (*Figure 2—figure supplement 4D*). Overall, a strong agreement was observed between datasets with UL83 contributing the majority of the protein, however, our data showed a greater enrichment of MCP, UL80, UL94 and gH (*Figure 2—figure supplement 4D*).

Collectively, we established that HCMV infection caused cells to release exosomes with increased density, that contain a subset of viral proteins different to the major virion constituents. HCMV infected cells also released virions that contained host proteins that were normally constituents of exosomes from uninfected cells. Therefore, a potential cross pollination between these two cellular processes continued to emerge.

## Uninfected cell exosomes, viral exosomes, and virions contain distinct protein profiles

Enrichment analysis and heatmap visualization revealed that both uninfected and infected gradients contained at least two vesicle/particle populations each. Due to incomplete isolation within a single gradient fraction, we pooled some neighboring fractions, and defined the four populations as exosome or MV (uninfected gradient), and viral exosome or virion (infected gradient) (*Figure 2G*). As we could not rule out that proteins in fraction one were not from lysed vesicles that consequently did not enter the gradient, this fraction was omitted from pooling. In addition, we realized that both MVs and virions resolved in fraction 7, and this may confound the analysis between exosomes and virions. So, we performed a quantitative comparison between exosomes and MVs (Comparison A, *Figure 2G*), and excluded any proteins from the exosome population that had less than 2-fold enrichment in exosomes compared to MVs (*Figure 2—figure supplement 4E*). This more stringently refined exosome proteome contained 765 proteins compared to 1225 in the MV, with 679 common to both populations (*Figure 2—figure supplement 4F*). The 679 proteins in the intersection accounted for the majority of total exosome protein, but strikingly, only contributed 7.4% of the total MV protein (*Figure 2—figure supplement 4G*). This indicated that these highly abundant exosome proteins are largely absent from MVs, and therefore the refined exosome proteome was used for all subsequent comparisons.

Pearson correlation and hierarchical clustering demonstrated strong reproducibility between biological triplicates, and revealed that the four populations were clearly distinct in their overall protein composition (*Figure 3A*). These observations were further supported by principal component analysis (*Figure 3B*). As somewhat expected, exosomes and viral exosomes were most similar to each other, and this pair was clearly distinguishable from virions. MVs were the least similar to any of the other populations. With four clearly distinct populations, we proceeded with quantitative comparisons focussing on addressing two key questions: (1) how does HCMV alter the proteome of exosomes released during infection? and (2) which host exosome proteins are integrated into HCMV virions?

## Viral exosomes contain specific HCMV proteins with unknown functions

HCMV significantly remodelled the protein composition of exosomes released during infection (*Figure 3A–B*). Viral exosomes contained 1647 host and 60 viral proteins (*Figure 3—figure supplement 1A–B*), comprising 97% and 3% of the total protein abundance, respectively (*Figure 3—figure supplement 1C*). To further investigate the specific enrichment of viral proteins in viral exosomes, and eliminate possible contamination from tailing or unresolved virions on the gradient, we first normalized the abundance of each viral protein within a population, and then calculated a ratio between populations (Comparison B, *Figure 2G*). As expected, abundant structural virion proteins UL83 and MCP had a ratio of 0.46 and 0.66 respectively (*Figure 3C*), indicating that they were more abundant in the virion. By contrast, envelopment and egress associated tegument proteins UL71, UL94 and UL103 (*Ahlqvist and Mocarski, 2011*; *Phillips and Bresnahan, 2012*; *Schauflinger et al., 2011*; *Womack and Shenk, 2010*) were approximately 6-fold enriched in viral exosomes, relative to virions (*Figure 3C*). However, the salient feature of this analysis was the identification of the viral Fc-gamma receptor homologue IR11/gp34 (*Corrales-Aguilar et al., 2014*), and uncharacterized transmembrane proteins US14 and IRL12 in the viral exosome (*Figure 3C*, *Figure 2—figure supplement 4C*). These proteins were enriched between 18–25-fold in the viral exosome, compared to the virion, and it is likely that they are embedded within the exosome membrane or specifically incorporated as molecular cargo. Moreover, this enrichment and unique localization may highlight a novel immuno-modulatory function of HCMV-induced exosomes, that has not yet been described, but currently outside the focus of our study.

## Uninfected cell exosomes and viral exosomes contain a similar host protein signature, but differential expression

Next, we compared the host proteins in exosomes and viral exosomes (Comparison C, *Figure 2G*). Of the 765 proteins identified in exosomes, only 11 were unique and absent from viral exosomes (*Figure 3D*). These 11 proteins comprised a negligible amount of the total protein abundance in exosomes (*Figure 3E*). Interestingly, the remaining 754 proteins were also identified in viral

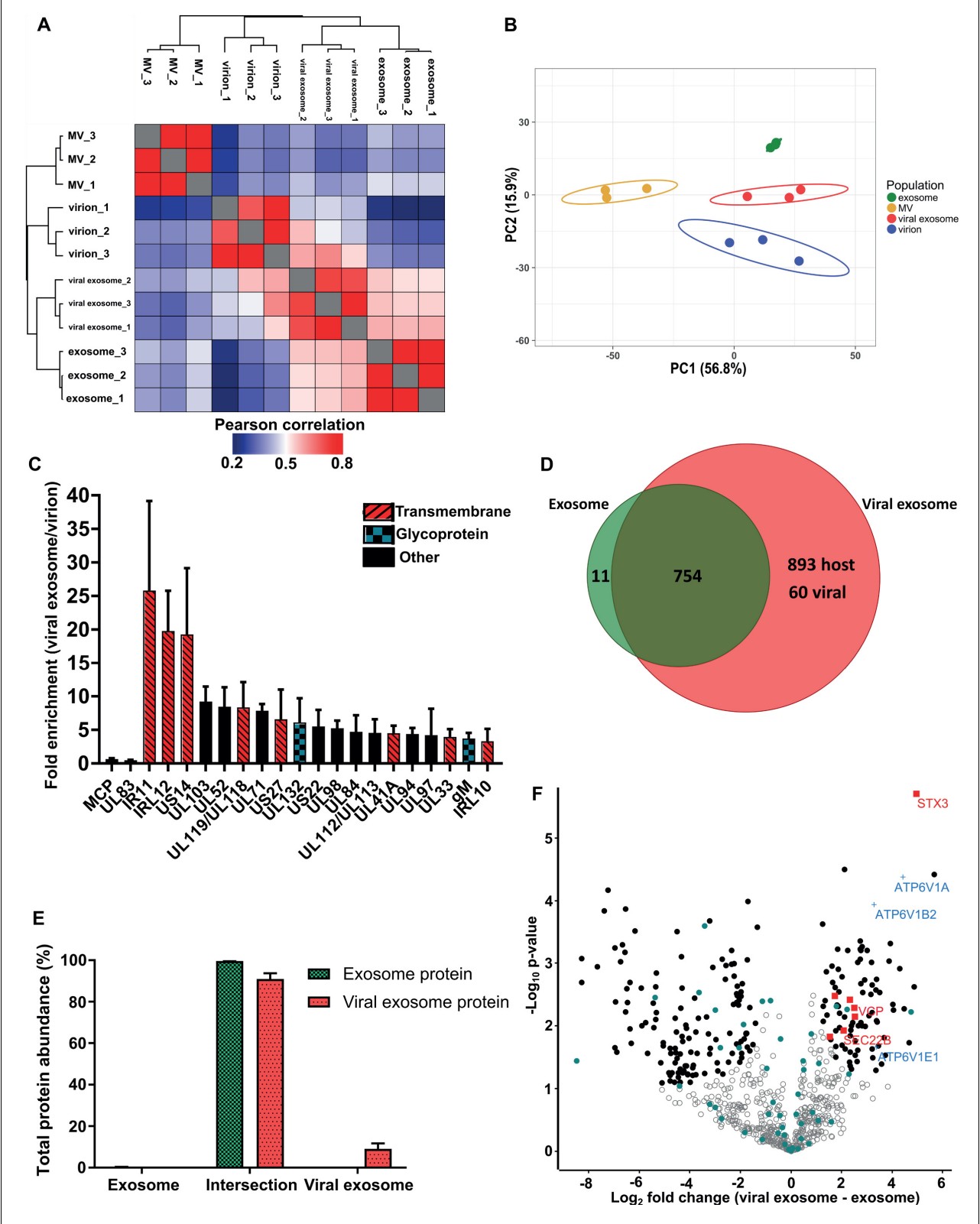

**Figure 3.** HCMV induces release of viral exosomes with remodelled host and viral cargo. (A) Comparison of protein profiles from all vesicle populations and biological replicates, as correlated by Pearson coefficient analysis. Populations were hierarchically clustered using k-means. (B) Comparison of vesicle population protein signatures by principle component analysis. Individual points plotted represent biological replicates, and circles represent 95% confidence intervals. (C) Histogram showing the relative enrichment of viral proteins in viral exosomes, compared to virions (n = 3,

*Figure 3 continued on next page*

*Figure 3 continued*

bars = SEM). Red hatching: predicted TMHMM transmembrane domain, solid black: no predicted TM domain, blue squares: viral glycoprotein. (D) Venn diagram depicting the total number, and proteins commonly identified in exosomes and viral exosomes. (E) Abundance of individual protein groups (based on Venn diagram in panel D), as a proportion of overall total protein abundance, in exosomes or viral exosomes (n = 3, bars = SEM). (F) Volcano plot showing relative enrichment of host proteins in viral exosomes (positive fold change), compared to exosomes (negative fold change). Red squares: envelopment/egress candidates, Blue crosses: V-ATPase sub-units, Teal circles: classical exosome markers, Black circles: significant differential expression (n = 3, $S_0$ = 0.4, FDR < 0.05, fold change >2).

The online version of this article includes the following figure supplement(s) for figure 3:

**Figure supplement 1.** Characterization of viral exosomes isolated on OptiPrep gradients.

exosomes, and constituted 91% of the total protein abundance of viral exosomes (*Figure 3E*). Although infection caused an additional 953 proteins to be identified in the viral exosome (*Figure 3D*), these proteins accounted for less than 9% of the total protein abundance (3% viral and 6% host) (*Figure 3E*, *Figure 3—figure supplement 1C*). Therefore, the host proteins in uninfected cell exosomes and viral exosomes are largely the same with some additional low abundance proteins present in viral exosomes.

Despite a similar qualitative composition of host proteins present in both exosomes and viral exosomes, we next tested for quantitative expression differences. We calculated and plotted the relative protein abundance between samples (*Figure 3F*), and observed a fold-change distribution indicating that some proteins were statistically enriched in exosomes (black circles, negative fold change), many that were equally expressed (grey circles), and a subset of proteins that were enriched in viral exosomes (black circles, positive fold change). Focussing on the latter, we searched for UniProt functional annotation terms 'vesicle', 'membrane' and 'traffic' which identified STX3, VCP, TMED10, SEC22B, MYH9, IGF2R and VPS35 to be of interest (red squares, *Figure 3F*, *Figure 3—figure supplement 1D*). Interestingly, STX3 was previously shown to be important for HCMV maturation (*Cepeda and Fraile-Ramos, 2011*; *Giovannone et al., 2017*), whilst VCP came up in a preliminary screen to identify host factors important for HCMV replication, assembly, and egress (*McCormick et al., 2018*). In addition, the V-ATPase has previously been implicated in HCMV progression (*McCormick et al., 2018*; *Pavelin et al., 2017*), and we observed that three sub-units of the V1 domain had increased expression in viral exosomes (blue crosses, *Figure 3F*), compared to uninfected exosomes. Taken together, this analysis discovered a cluster of host exosome proteins with implications in virion maturation, that were enriched in viral exosomes, relative to uninfected exosomes.

## Virions contain the complete repertoire of uninfected host exosome proteins, with a novel subset enriched

Examination of the virion proteome revealed 64 viral and 1712 host proteins (*Figure 4A*), with these proteins providing 36% and 64% of the total virion protein abundance, respectively (*Figure 4—figure supplement 1A*). When plotted against the ICL, virions isolated on OptiPrep gradients showed a very similar enrichment pattern of host organelle markers and viral proteins to the glycerol-tartrate purified virions (*Figure 4—figure supplement 1B*, *Figure 1C and H*). Exosome features within virions were further illuminated via direct comparison of the protein signatures (Comparison D, *Figure 2G*). Of the 765 proteins identified in exosomes, only 31 were not present in virions (*Figure 4A*). These proteins comprised only 0.6% of the total protein abundance, whilst the remaining 734 proteins represented 99.4% (*Figure 4B*). Importantly, these 734 exosome proteins were all subsequently integrated into the virion during infection (*Figure 4A*). This result indicated near-complete acquisition of the host exosome cargo during infection, and suggested that HCMV and exosomes bud from the same membrane, and are likely to use the same export mechanism.

To highlight host proteins incorporated into HCMV virions that may perform functional roles in virion maturation, we prioritized candidates based on an enrichment in virions, compared to uninfected exosomes. We first examined the 50 canonical exosome-associated markers (listed in *Figure 2A–B*). 45 of these were strictly classified as exosomal (based on enrichment comparisons with MVs, *Figure 2—figure supplement 4E*), and of these, 26 were significantly enriched in exosomes, 18 had no statistical change and only HSP90AA1 was found to be statistically enriched in

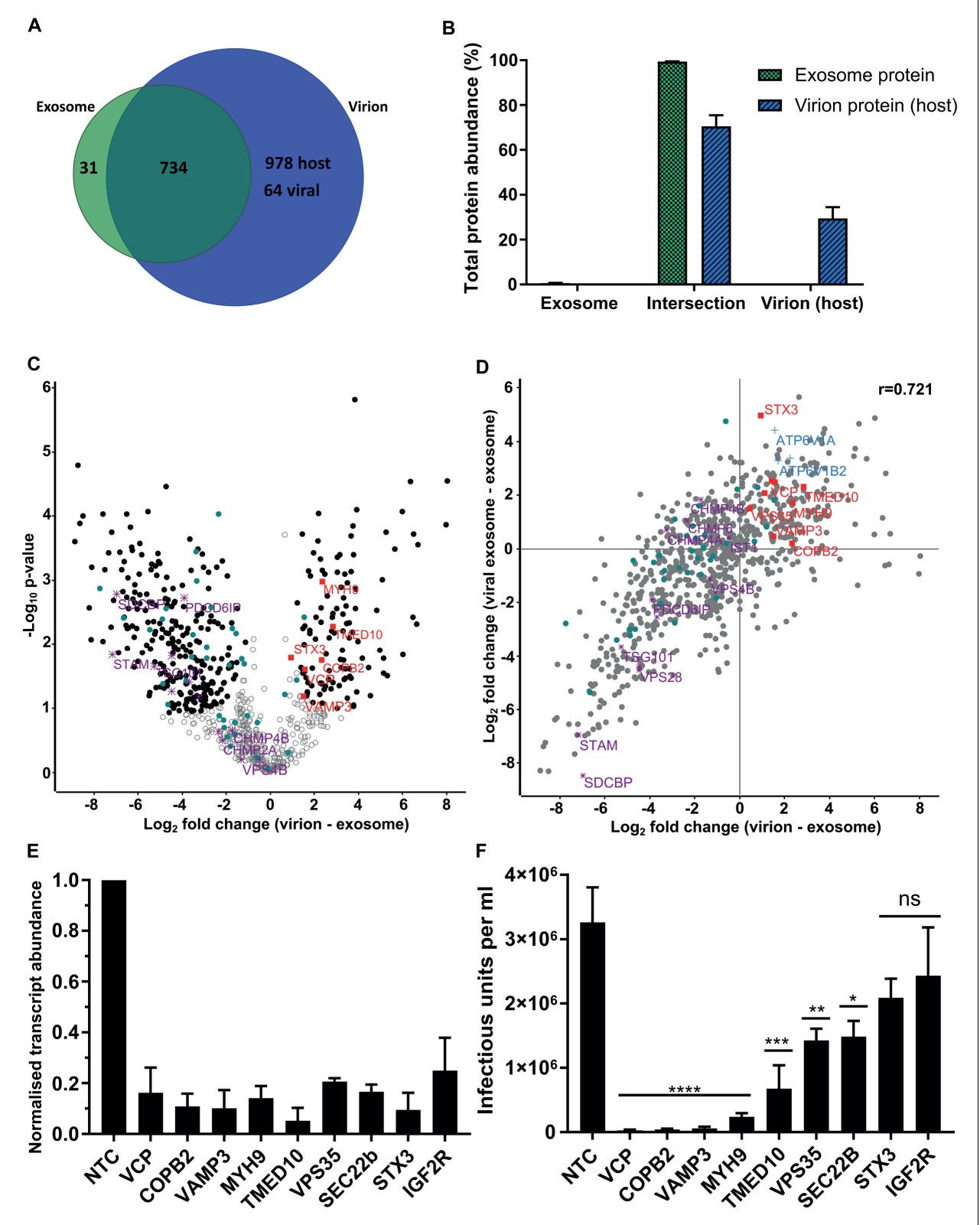

**Figure 4.** Host proteins in HCMV virions are predominantly exosome constituents. (**A**) Venn diagram depicting the total number, and proteins commonly identified in exosomes and virions. (**B**) Abundance of individual protein groups (based on Venn diagram in panel **A**), as a proportion of overall total protein abundance, in exosomes or virion host proteins (n = 3, bars = SEM). (**C**) Volcano plot showing relative enrichment of host proteins in virions (positive fold change), compared to exosomes (negative fold change). Black circles: significant differential expression (n = 3, $S_0 = 0.4$,

*Figure 4 continued on next page*

*Figure 4 continued*

FDR < 0.05, fold change >2). Red squares: envelopment/egress candidates, Teal circles: classical exosome markers, Purple stars: ESCRT sub-units. (D) Scatter plot correlating relative fold change comparisons of exosome vs viral exosome (*Figure 3F*), and exosome vs virion (C). Red squares: envelopment/egress candidates, Blue crosses: V-ATPase sub-units, Teal circles: classical exosome markers, Purple stars: ESCRT sub-units, (n = 3). (E) Relative transcript levels of host proteins following siRNA-mediated knock-down and HCMV infection (5 DPI, MOI = 3), normalized to cells treated with non-targeting control siRNA (n = 3, bars = SEM). (F) Quantification of infectious units released from siRNA-treated cells infected with HCMV AD169 5 DPI, MOI = 3. Infectious units in the supernatant were transferred to a reporter plate of uninfected cells, and IE1 assay conducted 24 HPI (n = 3, bars = SEM, ns: not significant, *p<0.05, **p<0.01, ***p<0.001, ****p<0.0001, one-way ANOVA with Dunnet's post-test).

The online version of this article includes the following figure supplement(s) for figure 4:

**Figure supplement 1.** Characterization of HCMV virions isolated on OptiPrep gradients.

virions (teal circles, *Figure 4C*). Next, we examined the relative expression of ESCRT proteins between uninfected cell exosomes and virions. Although all ESCRT-0 and ESCRT-I subunits were detected in exosomes and virions, none were significantly enriched, and SDCBP, STAM, TSG101 and Alix/PDCD6IP were among the least enriched exosome proteins in virions (purple stars, *Figure 4C*, *Figure 4—figure supplement 1C*). Similarly, ESCRT-III components and VPS4B had no significant change (*Figure 4C*).

Given the classical exosome markers showed no enrichment in virions, we instead focussed on the 112 host exosome proteins that significantly did (black circles, positive fold change, *Figure 4C*). We plotted fold change comparisons (exosome vs virion, and exosome vs viral exosome), and observed a strong positive correlation based on a Pearson coefficient of 0.721 (*Figure 4D*). Using this analysis, the 194 proteins in the top right quadrant (*Figure 4D*) were further queried for vesicle biogenesis or trafficking annotation, and nine candidates shortlisted to further investigate an involvement in producing infectious virions.

## Host exosome protein depletion inhibits HCMV replication

To examine an involvement producing infectious progeny, the expression of 9 host proteins were knocked-down (KD) using siRNA, and cells subsequently infected with HCMV. 5 DPI, we confirmed successful KD of target host genes by 75–95%, relative to the corresponding level in cells treated with non-targeting control (NTC) siRNA (*Figure 4E*). Simultaneously, cell supernatant from the infection was transferred to a reporter plate of uninfected cells, and IE1-positive cells counted 24 HPI. KD of all nine host proteins caused a decrease in the release of infectious virus, compared to the NTC (*Figure 4F*). However, IGF2R/CI-M6PR and STX3 KD did not achieve statistical significance, despite STX3 being previously reported to significantly reduce viral titre (*Cepeda and Fraile-Ramos, 2011*). KD of the other seven host proteins reduced virus production by varying orders of magnitude, with VCP, VAMP3, TMED10, MYH9 and COPB2 displaying the greatest inhibition (*Figure 4F*). Given the latter two proteins had been implicated in HCMV infection previously (*Jean Beltran et al., 2016*; *McCormick et al., 2018*), we instead focussed on VCP, TMED10, and VAMP3. Further assessment of cells subjected to KD conditions and low MOI infection revealed cells with KD VCP or TMED10 displayed defective cell proliferation and altered cell morphology (*Figure 5—figure supplement 1A*). This indicated that KD may have introduced a cellular toxicity that contributed to the cells inability to sustain HCMV production. In comparison, VAMP3 KD cells displayed no observable differences morphologically, and mimicked the NTC cells ability to form a coherent monolayer (*Figure 5—figure supplement 1A*). On this basis, the role of VAMP3 during HCMV infection was explored further.

## VAMP3 is essential for trafficking and release of infectious HCMV progeny

VAMP3 KD significantly reduced the ability of fibroblasts to produce AD169 HCMV (55-fold reduction compared to the NTC, *Figure 4F*). Conversely, fibroblasts over-expressing VAMP3 caused a significant increase in the release of AD169, compared to control cells (*Figure 5A*). The requirement for VAMP3 was not HCMV strain-specific, as production of low passage clinical isolate Merlin (RCMV1158) was blocked with even greater potency in VAMP3 KD cells (5200-fold reduction compared to the NTC, *Figure 5B*). We also measured viral spread in ARPE-19 cells using an epithelial-

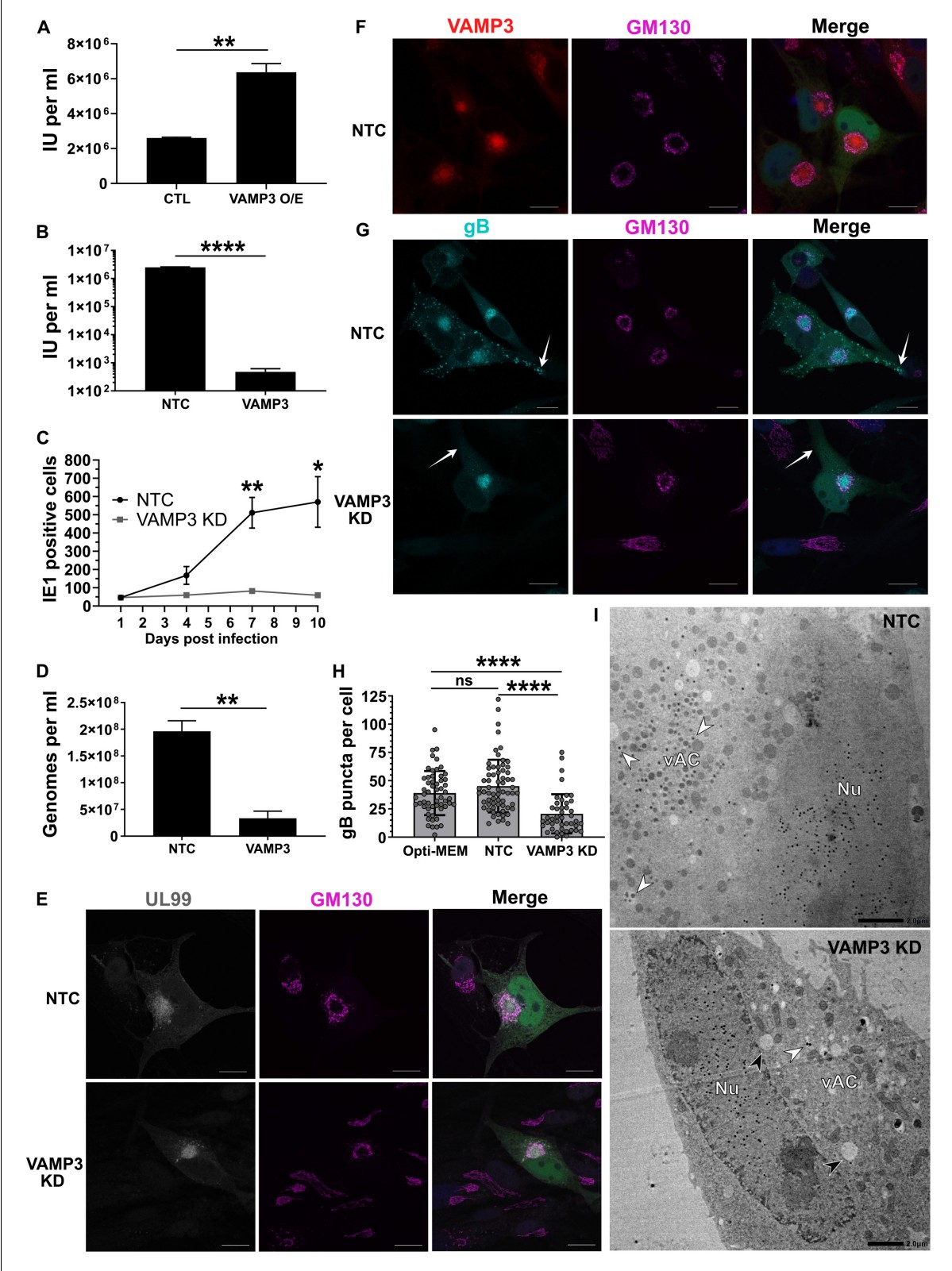

**Figure 5.** Host VAMP3 is essential for release of infectious HCMV progeny. (**A**) Quantification of infectious units released from fibroblasts over-expressing (O/E) VAMP3 or GFP-FLAG (CTL) fusion proteins. Cells were infected with HCMV AD169 MOI = 3, and 5 DPI the supernatant was transferred to a reporter plate of uninfected cells, and IE1 assay conducted 24 HPI. (n = 3, bars = SEM, **p<0.01, Student's t-test). (**B**) Quantification of infectious units released from siRNA-treated cells infected with HCMV strain Merlin (RCMV1158) MOI = 3. 7 DPI supernatant was transferred to a

*Figure 5 continued on next page*

*Figure 5 continued*

reporter plate of uninfected cells, and IE1 assay conducted 24 HPI (n = 3, bars = SEM, ****p<0.0001, Student's t-test). (**C**) Quantitation of HCMV spread in ARPE-19 cells treated with siRNA. Cells were infected with Merlin (RCMV1120) at low MOI. At indicated time-points post infection, cells were fixed directly and IE1 positive nuclei quantified (n = 3, bars = SEM, *p<0.05, **p<0.01, Student's t-test). (**D**) Quantitation of extracellular viral genomes in supernatant from siRNA treated cells infected with HCMV AD169, 5 DPI, MOI = 3. (n = 3, bars = SEM, **p<0.01, Student's t-test). (**E–G**) Immunofluorescence staining of UL99, GM130, VAMP3, and gB in siRNA-treated cells infected with HCMV-GFP AD169, 4 DPI, MOI = 0.1. White arrows: peripheral gB puncta, Scale bars = 20 µm. (**H**) Quantification of peripheral gB-positive puncta (outside Golgi ring) in siRNA-treated or Opti-MEM control cells, infected with HCMV-GFP AD169, 4 DPI, MOI = 0.1. Between 44 and 65 cells were quantified in each condition (n = 3, bars = SD, ns: not significant, ****p<0.0001, one-way ANOVA with Tukey's post-test). (**I**) Electron micrographs of VAMP3 siRNA and non-targeting control (NTC) treated cells infected with HCMV AD169, 5 DPI, MOI = 1. White arrows: maturing virions, Black arrows: enlarged empty vesicles, Scale bars = 2.0 µm.

The online version of this article includes the following figure supplement(s) for figure 5:

**Figure supplement 1.** Involvement of VAMP3 during HCMV infection.

specific Merlin virus (RCMV1120), with a low MOI infection, over 10 days. Similarly, generation of IE1-positive cells was reduced by approximately 10-fold at 10 DPI in VAMP3 KD cells (*Figure 5C*). Therefore, VAMP3 is an essential host protein required for HCMV replication across multiple virus strains, and cell types.

To determine the maturation stage that VAMP3 was required, we first tested whether viral genomes were replicated in the absence of VAMP3. Quantification revealed robust genome replication in VAMP3 KD cells, and a small relative reduction (1.7 fold) compared to NTC cells (*Figure 5— figure supplement 1B*). Assessment of late-stage viral proteins UL99, UL83, gB and MCP revealed that VAMP3 KD cells had similar expression levels as NTC cells (*Figure 5—figure supplement 1C*), indicating no obvious delay in viral gene expression. So next we investigated the involvement of VAMP3 in the context of vAC biogenesis, virion cargo recruitment, secondary envelopment, egress, and virion re-entry.

We investigated whether the reduced infectious units derived from VAMP3 KD cells (*Figure 4F*), was due to a primary infection defect (i.e. lower number of total virions released by VAMP3 KD cells) or a secondary infection defect (i.e. same number of virions released compared to NTC, but defective virions that failed to re-infect the reporter plate). The total number of HCMV genome copies released from infected cells into the conditioned medium was quantified, and confirmed that significantly less viral genomes were present in the supernatant of VAMP3 KD cells, compared to NTC cells (*Figure 5D*). This indicated that VAMP3 function was required in the primary cell to release infectious HCMV virions.

As vAC generation is essential for production of infectious progeny (*Das et al., 2014*), we used this HCMV hallmark as a molecular checkpoint to assess infection progression and function in VAMP3 KD cells. Similar to NTC cells, infected VAMP3 KD cells exhibited a prototypical vAC consisting of a GM130-positive Golgi ring, clustered around the viral tegument protein UL99 (*Figure 5E*). In addition, VAMP3 accumulated within the Golgi ring of the vAC in NTC cells (*Figure 5F*), and this localization is associated with virion maturation (*Silva et al., 2003*; *Liu et al., 2011*). On the basis that VAMP3 is a v-SNARE involved in vesicular transport (*Hu et al., 2007*), viral trafficking was investigated next.

Monitoring the localization of the virion envelope protein gB during wild-type infection revealed an accumulation in the vAC by 72 HPI (*Figure 5—figure supplement 1D*). This was followed by the emergence of numerous brightly-stained puncta (approximately 1 µm in diameter) towards the cell periphery by 96 HPI (*Figure 5—figure supplement 1D*), likely representing egressing virions (*Goulidaki et al., 2015*; *Sampaio et al., 2005*). In NTC cells at 96 HPI, both VAMP3 and gB accumulated within the vAC (*Figure 5—figure supplement 1E*), and gB puncta were dispersed throughout the cytoplasm, including within cell projections (white arrows, *Figure 5G*). Strikingly, no cytoplasmic gB puncta were observed outside the Golgi ring in VAMP KD cells (*Figure 5G*), and cellular quantification further highlighted their significant loss (*Figure 5H*). This indicated a lack of maturing virions in the cytoplasm and cell periphery, and to confirm this we performed EM analysis. Infected NTC cells showed capsids in the nucleus, and a clear accumulation of virions clustered in the vAC (white arrows, *Figure 5I*). In comparison, although VAMP3 KD cells had capsids in the nucleus, the vAC lacked any virion maturation activity (*Figure 5I*). Instead, numerous large, empty vacuoles were

present (black arrows, *Figure 5I*). Taken together, we concluded that VAMP3 is an essential host protein required for traffic of virion cargo to the vAC for subsequent virion maturation.

## Discussion

Due to the complexity, defining the molecular process by which HCMV orchestrates virion assembly and egress has been challenging and enigmatic. We targeted the dependency of HCMV upon the host cell architecture to uncover novel insights into virion maturation. By analyzing extracellular virions as the end product, we reverse-engineered the system and discovered a large proportion of host exosome protein constituents. The parallels that exist between the exosome biogenesis pathway and virion assembly and egress is striking. Both exosomes and virions are derived from the inward budding of endosomal membranes, require trafficking of MVBs to the cell surface, and require fusion events with the plasma membrane for extracellular release (*Figure 6*). We reasoned that the ability of HCMV to exploit this pre-existing host pipeline would promote completion of its replication cycle, and is consistent with other herpesviruses including Human Herpes Virus 6 (*Mori et al., 2008*), and Herpes Simplex Virus 1 (*Crump et al., 2007*; *Pawliczek and Crump, 2009*; *Calistri et al., 2007*) that have previously implicated exosome proteins in their virion egress. Therefore, our discovery that HCMV exploits the host exosome pathway using a defined set of non-classical molecular effectors, advances knowledge to better understand the cellular processes that underpin virion maturation.

The complexity of vesicles and particles released from infected cells is challenging, with various populations present in conditioned medium. We confirmed that the abundance of DBs in virion preps can be minimised by omitting sonicated cell supernatant as input for gradient purification. Overall, classical glycerol-tartrate gradients were effective at separating NIEPs, virions, DBs and cellular vesicles. Although we only analyzed virions from these gradients, future studies will define the host proteomic signatures in the other species. Similar to glycerol-tartrate gradients, Optiprep gradients effectively separated viral exosomes, virions, and DBs. Whilst we did not discriminate NIEPs on these gradients, we do not expect their presence to impact our virion membrane findings since the only difference between virions and NIEPs is the lack of a viral genome in the capsid of the latter.

In addition to virions, infected cells released viral exosomes with a unique protein composition. We are confident NIEPs did not contribute to the altered protein profile of viral exosomes, as no capsid containing particles were observed by EM in fraction 3 (*Figure 2F*), and structural capsid proteins were only detected at very low levels in these fractions (*Figure 3C*, *Supplementary file 2*). Interestingly, during completion of our study, a FACS-based experiment reported that HCMV infected cells released extracellular vesicles positive for viral envelope proteins gB and gH (*Zicari et al., 2018*). Whilst our findings also support this observation (*Figure 2C*), our proteomic approach provided considerably greater depth, and enabled enrichment calculations for all proteins in virions versus viral exosomes. Analysis of the 64 viral proteins revealed that compared to virions that contained all the classical viral tegument and envelope proteins, viral exosomes were instead enriched with the viral Fc-gamma receptor homologue IR11/gp34 (*Corrales-Aguilar et al., 2014*), and uncharacterized transmembrane proteins US14 (*Das and Pellett, 2007*) and IRL12. It is tempting to speculate that these viral proteins exported in exosomes from infected cells serve immuno-evasion functions. For example, IR11 may bind and sequester antibody, and compared to a cell surface localization, its exosome-based release would increase its range of decoy action. Interestingly, exosomes released from Epstein-Barr Virus have been well characterized, and are known to contain LMP1 that mediates immunosuppressive capacity (*Dukers et al., 2000*; *Flanagan et al., 2003*), whilst exosomes from Kaposi Sarcoma-associated Herpesvirus have been reported to have pleiotropic functions in the microenvironment (*Meckes et al., 2013*).

A striking feature of our study was the finding that virtually all proteins in uninfected exosomes were subsequently integrated into virions during infection (*Figure 4B*). This indicated that maturing nucleocapsids acquire their outer envelope from the same endosome/MVB membrane that ILVs/exosomes are derived from (*Figure 6*). In addition, although viral exosomes and virions have very different viral protein cargo (*Figure 2C*), they too share an almost identical host protein composition (*Figure 3—figure supplement 1A–B*). This similarity suggested a common membrane origin, and that viral exosomes and virions bud into the same MVB during infection (*Figure 6*). Although outside

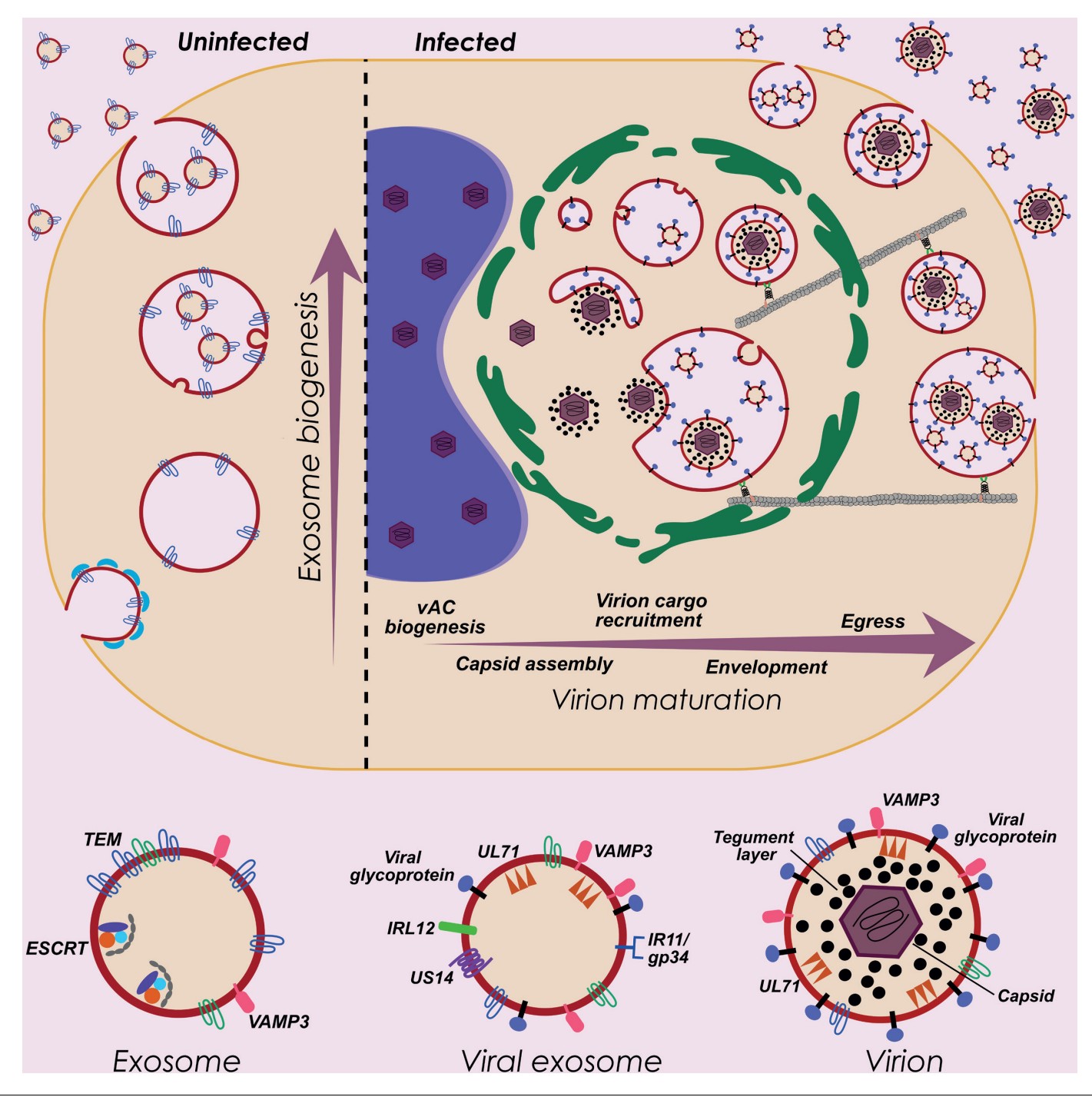

**Figure 6.** HCMV exploits host exosome biogenesis for virion assembly and egress. Proposed model describing key stages of virion maturation. HCMV induces vAC generation to prepare the host cell architecture for virion assembly and egress. Endosome-derived membranes/MVBs that normally give rise to exosomes in uninfected cells, are sequestered within the vAC during infection, and provide the membrane for viral exosomes and virions. Virion cargo and genome-containing nucleocapsids enter the vAC and acquire their outer envelope by budding into the host endosome-derived membranes/MVBs. Subsequent cellular trafficking and fusion enables virions to egress and be released from infected cells. Inhibition of key host or viral modulators at associated stages can block virion maturation.

the scope of this current study, we were intrigued by how the viral cargo specificity for each population could be achieved, and propose two alternate models. The first is based upon a temporal aspect and the long replicative cycle. Given host cells are continuously secreting exosomes, if viral exosome cargo (ie. IR11 et al.) were translated and incorporated into the (common) MVB earlier in infection, the biogenesis and release of viral exosomes would precede virions that access the MVB at a later stage. We were unable to distinguish this as we collected the infected cell supernatant at a single time so both populations were collected together. The second model is based upon viral exosomes and virions budding from the same MVB simultaneously. Therefore, their biogenesis might be dependent upon microdomains within the MVB membrane that allow clustering of specific receptors, or that enable unique protein-protein interactions to drive their differential envelopment. Confirmation of the precise mechanisms awaits future experiments.

As exosome cargo includes proteins responsible for their biogenesis and subsequent export, our discovery that all exosome proteins were incorporated into virions revealed a novel suite of 734 host proteins as candidate regulators of virion maturation. We acknowledged that many of these may not perform functional roles, and instead be co-opted into the virion by association/localization. So, we used quantitative criteria (*Figure 4D*) to prioritize a small subset of candidates for functional testing. This proved to be successful with KD of 7/9 host proteins significantly decreasing HCMV production (*Figure 4F*). Whilst we only selected nine proteins from the 194 that had a positive fold change in both comparisons and present in the top right quadrant (*Figure 4D*), it is likely that several others will also impact production of HCMV virions. Nonetheless, a quantitative difference does not always indicate a functional involvement, and we predict that a proportion of the 734 proteins with no statistical virion enrichment, may already have sufficient physiological levels to functionally impact virion maturation. In addition, we expect that other host proteins that lack a virion cargo fate are also able to modulate virion packaging and release. As such, complementary reasoning, criteria, and orthogonal approaches will be required for their discovery.

We found that VAMP3 KD cells were able to generate a vAC that clustered UL99 and gB, before the replication cycle stalled. Interestingly, after completion of experiments for this manuscript, it was shown that KD of VAMP3 in hippocampal neurons significantly inhibited exosome release (*Kumar et al., 2020*). This result further supports our conclusion that host proteins required for exosome biogenesis are also essential for HCMV maturation. EM analysis revealed that although the vAC was present in VAMP3 KD cells, it was completely devoid of maturing virions. This indicates that VAMP3 is required for virion cargo recruitment into the vAC, and is consistent with its cellular function as a v-SNARE that mediates intracellular vesicle traffic by fusion with its corresponding t-SNARE (*Hu et al., 2007*; *Puri et al., 2013*). Interestingly, SNAP-23 is a t-SNARE for VAMP3, and has also been shown to localize within the vAC during infection (*Liu et al., 2011*). Therefore, VAMP3 may deliver its cargo into the vAC for subsequent maturation. In the absence of VAMP3, maturing virions did not accumulate in the vAC, and consequently, no gB trafficking was observed, and no infectious virus released. This highlights virion cargo recruitment as an important stage of the egress continuum (*Figure 6*), and that blocking virion cargo traffic may confer an antiviral advantage. Given VAMP3 knock-out mice survive and appear to be phenotypically normal (*Yang et al., 2001*), the inhibition of VAMP3 may not severely compromise uninfected cells, but abolish virion egress in infected cells. This selectivity further supports the future investigation of VAMP3-dependent viral trafficking from a therapeutic perspective.

In this study, VAMP3 served exemplar to validate our discovery that host exosome proteins perform essential functions during HCMV virion maturation. This finding now pinpoints a molecular axis to direct future investigations to further illuminate virion assembly and egress. Given the continuum of cellular processes involved in producing infectious virus, it is likely that a diverse set of host proteins confer specific functions at key replication checkpoints. Therefore, our datasets serve as a key resource that can be broadly utilised to define critical host proteins involved in vAC generation, cargo recruitment, secondary envelopment, virion trafficking, MVB fusion and extracellular release (*Figure 6*). Collectively, the knowledge revealed by our study significantly enhances the understanding of viral egress, promotes its antiviral development in the future, and has the potential to impact the maturation of other herpesviruses.

# Materials and methods

## Key resources table

| Reagent type (species) or resource | Designation | Source or reference | Identifiers | Additional information |
|---|---|---|---|---|
| Cell line (*Homo-sapiens*) | MRC5 primary fetal lung fibroblasts | Purchased from ATCC | CCL-17 | |
| Cell line (*Homo-sapiens*) | ARPE-19 retinal pigment epithelial cells | Purchased from ATCC | CRL-2302 | |
| Cell line (*Homo-sapiens*) | Phoenix-AMPHO kidney epithelial cells | Purchased from ATCC | CRL-3213 | |
| Strain, strain background (*Human Cytomegalovirus*) | HCMV BAC clone | Thomas Shenk (Princeton University) (*Yu et al., 2002*) | AD169 | |
| Strain, strain background (*Human Cytomegalovirus*) | HCMV BAC clone | Richard Stanton (Cardiff University) (*Stanton et al., 2010*) | RCMV1158 | Unrepaired Merlin virus with GFP cassette |
| Strain, strain background (*Human Cytomegalovirus*) | HCMV BAC clone | Richard Stanton (Cardiff University) (*Stanton et al., 2010*) | RCMV1120 | UL128 repaired Merlin virus without tags |
| Recombinant DNA reagent | pLXSN | Clontech | 631509 | Retroviral construct to transduce and express VAMP3 |
| Sequence-based reagent | vamp3_F | This paper | PCR primer | 5'-TGACGAATTCATG TCTACAGGTCCA-3' |
| Sequence-based reagent | vamp3_R | This paper | PCR primer | 5'-TGAGGATCCTCATG AAGAGACAAC-3' |
| Sequence-based reagent | ul83_F | *Gault et al., 2001* | qPCR primer | 5'-GTCAGCGTTC GTGTTTCCCA-3' |
| Sequence-based reagent | ul83_R | *Gault et al., 2001* | qPCR primer | 5'-GGGACACAACA CCGTAAAGC-3' |
| Sequence-based reagent | atg5_F | This paper | qPCR primer | 5'-TATTACCCTTTGA TGCCTTTTTTTC-3' |
| Sequence-based reagent | atg5_R | This paper | qPCR primer | 5'-AACTTGTACCAC CAATTCTAAAATG-3' |
| Sequence-based reagent | siRNA screen qPCR primers | This paper | | All sequences available in *Supplementary file 3* |
| Sequence-based reagent | siGENOME SMARTpool siRNA | Dharmacon | All catalogue numbers available in *Supplementary file 3* | All sequences available in *Supplementary file 3* |
| Antibody | anti-human CD63 (Mouse monoclonal) | Santa Cruz | sc-5275 | WB (1:1000) |
| Antibody | anti-human CD81 (Mouse monoclonal) | Santa Cruz | sc-166029 | WB (1:1000) |

*Continued on next page*

*Continued*

| Reagent type (species) or resource | Designation | Source or reference | Identifiers | Additional information |
|---|---|---|---|---|
| Antibody | anti-human VPS4 (Mouse monoclonal) | Sigma | SAB4200215 | WB (1:1000) |
| Antibody | anti-human TSG101 (Mouse monoclonal) | GeneTex | GTX70255 | WB (1:1000) |
| Antibody | anti-human ALIX/PDCD6IP (Mouse monoclonal) | Cell Signalling Technology | #2171 | WB (1:1000) |
| Antibody | anti-HCMV gB (Mouse monoclonal) | Abcam | ab6499 | WB (1:1000) IF (1:500) |
| Antibody | anti-HCMV UL83 (Mouse monoclonal) | Thomas Shenk (Princeton University) *Nowak et al., 1984* | Clone 8F5 | WB (1:1000) |
| Antibody | anti-HCMV UL99 (Mouse monoclonal) | Thomas Shenk (Princeton University) *Silva et al., 2003* | Clone 10B4 | WB (1:1000) IF (1:500) |
| Antibody | anti-human β-actin (Mouse monoclonal) | Sigma | A2228 | WB (1:1000) |
| Antibody | anti-human VAMP3 (Rabbit monoclonal) | Abcam | ab200657 | WB (1:1000) IF (1:500) |
| Antibody | anti-HCMV IE1 (Mouse monoclonal) | Thomas Shenk (Princeton University) (*Zhu et al., 1995*) | Clone 1B12 | IF (1:100) |
| Antibody | anti-human GM130 (Mouse monoclonal) | BD Biosciences | 610822 | IF (1:500) |
| Antibody | anti-human GM130 (Rabbit monoclonal) | Abcam | ab52649 | IF (1:500) |
| Antibody | goat anti-mouse Alexa Fluor (568) | Invitrogen | A-11004 | IF (1:1000) |
| Antibody | goat anti-rabbit Alexa Fluor (633) | Invitrogen | A-21070 | IF (1:1000) |
| Antibody | goat anti-mouse HRP conjugate | Bio-Rad | #1706516 | WB (1:5000) |
| Antibody | goat anti-rabbit HRP conjugate | Bio-Rad | #1706515 | WB (1:5000) |
| Commercial assay or kit | Viral DNA spin kit | Qiagen | Cat No./ID: 57704 | |
| Software, algorithm | MaxQuant | MaxQuant (*Cox and Mann, 2008*) | version 1.6.0.13 | |
| Software, algorithm | Perseus | Perseus (*Tyanova et al., 2016*) | version 1.6.0.7 | |
| Software, algorithm | GOPlot | GOPlot (*Walter et al., 2015*) | v.1.0.2, CRAN | |

## Cells and virus

MRC5 primary fetal lung fibroblasts (ATCC CCL-17) and ARPE-19 retinal pigment epithelial cells (ATCC CRL-2302) were purchased directly from ATCC, were grown in Gibco DMEM (MRC5) or Gibco DMEM/F-12 (ARPE-19) full media, supplemented with 10% (v/v) Fetal Bovine Serum (Cell

Sera), 10 U/ml penicillin and 10 U/ml streptomycin and cultured in 5% $CO_2$ at 37°C. All cells were regularly tested for mycoplasma contamination using PCR, and found to be negative. For vesicle-related experiments, confounding bovine vesicles were depleted from serum by centrifugation at 100,000xg for 1 hr at 4°C.

HCMV was reconstituted by electroporating MRC5 or ARPE-19 cells with bacterial artificial chromosomes (BAC) containing the AD169 genome kindly provided by Prof. Thomas Shenk (*Yu et al., 2002*), or Merlin kindly provided by Dr Richard Stanton (*Stanton et al., 2010*). Cellular supernatant was harvested and a sorbitol cushion (20% (v/v) D-sortibol, 50 mM Tris pH 7.4) underlaid, prior to ultracentrifugation at 50,000xg for 1 hr at 4°C (*Britt, 2010*). Virus pellets were resuspended in full media, titred by TCID$_{50}$ assay, and stored at −80°C. For HCMV infections, virus was added to cells at the specified multiplicity of infection (MOI) with gentle agitation over 3 hr, then aspirated and replaced with full media until harvesting.

## HCMV virion purification and proteinase K surface shaving

Fibroblasts were cultured in 6–8 roller bottles, and medium replaced with serum-free DMEM containing pen/strep prior to infection. Cells were infected (MOI = 0.01) with BAC-derived AD169, and cell supernatant collected when 100% of cells were infected before significant cell death. The supernatant was clarified by centrifugation at 1000x g for 10 min, and centrifuged through a 20% sorbitol cushion at 72,000x g for 1.5 hr. The pellet was resuspended in buffer containing 50 mM Tris pH7.4 and 100 mM NaCl (TN buffer) and centrifuged through a sodium tartrate-glycerol gradient to separate virions, non-infectious enveloped particles (NIEPs), and dense bodies from cell-derived vesicles. Individual bands were extracted, diluted 3-fold in TN buffer, and further isolated on a second gradient by centrifugation at 75,000x g for 1 hr. Virions were then either washed with PBS and analyzed by mass spectrometry, or digested with proteinase K. For the surface shaving experiments, virions were treated with 10 ug proteinase K for 3.5 hr. After digestion, virions were diluted 9-fold with PBS containing proteinase inhibitors (Roche), and isolated on another sodium tartrate-glycerol gradient. Finally, virions were extracted, washed with PBS, and analyzed by mass spectrometry.

## Extracellular vesicle separation

MRC5 cells were seeded into 150 mm dishes (5.5 $\times$ 10$^6$ cells per plate). four dishes were infected with HCMV at MOI = 3, and cellular supernatant collected 5 days post infection (DPI). Supernatant from 19 dishes of uninfected cells was also collected 5 days post mock infection. Supernatant collected from both conditions was first clarified at 1500xg for 10 min, then centrifuged at 100,000xg for 90 min at 4°C. Vesicle pellets were washed once in PBS, and resuspended in PBS to determine the protein concentration using the Bradford total protein reagent (BioRad), according to the manufacturer's instructions. Equal protein amounts from uninfected or infected vesicle samples were added to 5% Optiprep diluted in HEPES buffered saline (0.85% (w/v) NaCl, 10 mM HEPES-NaOH, pH 7.4) and layered onto continuous 10–40% Optiprep density gradients. Ultracentrifugation was performed at 140,000xg for 16 hr at 4°C, after which, 10 individual fractions were aspirated manually, washed with PBS, and used immediately in downstream experiments.

## Densitometry-based protein quantification

Vesicle pellets from individual fractions were lysed with 30 μl LDS buffer (141 mM Tris pH 8.5, 2% (w/v) LDS, 10% (v/v) glycerol, 0.51 mM EDTA, 0.22 mM G250, 50 mM DTT), and denatured at 95°C for 5 min. A 5 μl aliquot from every sample, and 5 μl BenchMark protein standard (Invitrogen) was separated by PAGE (NuPAGE 4–12% Bis-Tris gels), and the gels stained with SYPRO ruby (Thermo-Fisher Scientific) according to manufacturer's instructions. Gels were imaged using a Typhoon Trio (GE Healthcare) at a resolution of 200 μm, and excitation (470 nm) and emission filters (610 nm) set. The protein concentration in each fraction was determined by relative densitometry of each lane compared to the BenchMark standard, as calculated by ImageQuant analysis toolbox software (GE Healthcare).

## Western blotting

Samples were subjected to PAGE as above, and proteins wet transferred to PVDF membranes using a Mini-PROTEAN Tetra cell (BioRad) at 100 V for 60 min at 4°C. Membranes were blocked with 5%

(w/v) skim milk in 1X TBST (150 mM NaCl, 50 mM Tris pH 7.4, 0.1% (v/v) Tween 20) for a minimum of 2 hr at RT, and membranes incubated with primary antibodies: mouse anti-CD63 (Santa Cruz sc-5275), anti-CD81 (Santa Cruz sc-166029), anti-VPS4 (Sigma SAB4200215), anti-TSG101 (GeneTex GTX70255), anti-Alix/PDCD6IP (Cell Signalling Technology #2171), anti-HCMV gB (Abcam ab6499), anti-UL83 (Clone 8F5 *Nowak et al., 1984*), anti-UL99 (Clone 10B4 *Silva et al., 2003*), anti-β-actin (Sigma, A2228) and rabbit anti-VAMP3 (Abcam ab200657), diluted 1:1000 in 5% skim milk for 1 hr at RT, or overnight at 4°C. Membranes were washed with TBST, and appropriate HRP-conjugated secondary antibodies (Bio-Rad) incubated for 1 hr at RT. After washing in TBST, membranes were incubated in Clarity ECL substrate (Bio-Rad) for 1 min, and imaged using a Gel Doc imaging system (Bio-Rad). Images were viewed and analyzed using ImageJ.

### Infectivity assay of gradient fractions

Supernatant from approximately $1.8 \times 10^7$ infected cells was collected and fractionated as above. Instead of lysing vesicles for PAGE-related analysis, washed vesicle pellets in each fraction were resuspended in full media, serially diluted (1:10), and incubated with a 24-well reporter plate containing uninfected MRC5 cells. 1 DPI, cells were fixed in 2% (v/v) formaldehyde in PBS for 15 min at RT, permeabilized with 0.1% (v/v) Triton X-100 in PBS, and washed 3 times with 0.2% (v/v) Tween 20 in PBS. Cells were blocked at RT for 1 hr, incubated with mouse anti-IE1 primary antibody (1:100, Clone 1B12 *Zhu et al., 1995*) for 90 min, washed as above, and incubated with goat anti-mouse Alexa Fluor secondary antibody for 30 min (1:1000). 1% (v/v) Hoechst stain in PBS was incubated at 37°C for 10 min, and washed prior to imaging. 3 fields of view were imaged for each well with a Nikon eclipse TE2000-U confocal microscope. IE1 positive cells were counted manually.

### Proteomics

#### Tryptic protein digestion and peptide desalting

Protein samples were reduced and alkylated with 50 mM TCEP and 50 mM chloroacetamide, at 95°C for 10 min. Digestion was based on the FASP method (*Manza et al., 2005*; *Wiśniewski et al., 2009*), with slight modification. Samples were transferred to Millipore Ultra-0.5 10 kDa centrifugal filters (Amicon), and buffer-exchanged three times into TUD buffer (0.1 M Tris-HCl, 2% (w/v) sodium deoxycholate, 8 M urea), by centrifugation at 14,000xg for 10 min at RT. For the final exchange, protein samples in 200 μl TUD were mixed with 200 μl of ABC-DOC buffer (50 mM ammonium bicarbonate, 2% (w/v) sodium deoxycholate), and centrifuged as above. 100 μl of digestion buffer (50 mM ammonium bicarbonate), 0.5 μg MS grade trypsin (Pierce) was added, and incubated overnight at 37°C. Digested peptides were recovered, mixed with ethyl acetate (in 2:1 ratio), and adjusted to a final concentration of 0.5% (v/v) trifluoroacetic acid (TFA). Following vortexing for 2 min, samples were centrifuged at 14,000xg for 5 min at RT, and the denser aqueous phase recovered. Peptides were desalted on StageTips (*Rappsilber et al., 2003*) using Empore SDB-RPS extraction discs (3M Analytical Biotechnologies). Bound peptides were washed first with a solution containing 50% (v/v) ethyl acetate, 0.5% (v/v) TFA, and a second wash with 0.5% (v/v) TFA. Finally, peptides were eluted with 5% (v/v) ammonium hydroxide, 80% (v/v) acetonitrile, concentrated to near dryness using a vacuum centrifuge (CentriVap), and FA solution (0.1% (v/v) formic acid, 2% (v/v) acetonitrile) added to achieve a final volume of 9 μl. Samples were transferred to autosampler vials for liquid chromatography and mass spectrometry analysis (LC-MS).

#### Mass spectrometry and protein identification

LC-MS was performed using an UltiMate 3000 Ultra High-performance Liquid Chromatography system coupled to Orbitrap Fusion Tribrid mass spectrometer (ThermoFisher.Scientific). Sample peptides were trapped on an Acclaim PepMap 100 column (C18, 2 cm length, 100 μm ID, 5 μm particle size, 100 Å pore size; ThermoFisher Scientific), and subsequently loaded onto an Acclaim PepMap RSLC analytical column (C18, 50 cm length, 75 μm ID, 3 μm particle size, 100-Å pore size; ThermoFisher Scientific). Peptides were eluted at 250 nL/min, using a linear gradient over 90 min, starting at 7.5% Buffer B (80% acetonitrile, 0.1% formic acid) in Buffer A (0.1% formic acid), and ending at 37.5% Buffer B in Buffer A. The Fusion mass spectrometer was operated in data-dependent acquisition mode with a single acquisition cycle comprising one full-scan (m/z 375–1575) in the Orbitrap (resolution of 120,000), followed by higher-energy collisional dissociation (HCD) fragmentation of

the top 15 most intense precursor ions, (resolution of 30,000) using 32% collision energy. Dynamic exclusion was enabled for 15 s.

Raw spectral files were analyzed using MaxQuant (version 1.6.0.13 *Cox and Mann, 2008*). Peptides spectra were searched using the Andromeda search engine built into MaxQuant (*Cox et al., 2011*), against a combined Uniprot database comprising Human (Taxon ID: 9606, 20,432 entries) and HCMV strain AD169 (Taxon ID: 10360, 193 entries) proteins. Generally, MaxQuant default search parameters were used, with limited changes. Cysteine carbamidomethylation was set as a 'fixed modification', methionine oxidation and N-terminal acetylation were set as 'variable modifications', and trypsin/P selected as the 'digestion enzyme'. Peptide length was set between 7 and 25 residues with a maximum number of 2 missed cleavages allowed. 'Label-free quantification (LFQ)' and 'match between runs' were enabled, and protein false discovery rate set to one percent with a minimum number of 2 'razor and unique peptides' required for identification.

### Bioinformatics and statistical analysis

MaxQuant proteinGroups files were uploaded into Microsoft Excel for downstream data management, quality control, and normalization. Protein contaminants and reverse hits were deleted from the dataset, and LFQ intensity measurements used for quantitative comparisons in Perseus (*Tyanova et al., 2016*) to generate heatmaps and volcano plots. All LFQ values were $log_2$ transformed, missing values ($log_2(0)$=NaN) were randomly imputed from the whole matrix distribution after transforming by a shrink factor of 0.3 and down shifting 1.8 standard deviations. For curated exosome marker and viral protein heatmaps, the mean values for each fraction from the triplicate were taken, and subsequently Z-scored across each row. Hierarchical clustering of matrix rows was performed using Euclidean distance (average linkage, K-means) with number of clusters set to 15 with 10 iterations and one restart.

To define vesicle populations, mean LFQ values for each protein from pooled fractions (either 2 or three depending on population) was taken. The average for each protein was divided by the summed total of all averages within a defined population. The sum of all LFQ values for all proteins in each defined population is 1. Normalized values were $log_2$ transformed and matrix rows were further filtered by 'at least two valid values in at least one population' to reduce the number of imputed values. For volcano plots, missing values were imputed as before. Statistical significance was determined by the Student's t-test, with permutation based false discovery rate set to 0.05, $S_0$ constant of 0.4 and a fold change greater than 2. For PCA, missing values were imputed from individual matrix columns (shrink factor 0.3, downshift 1.8 std deviations). Analysis and plots were produced using ClustVis with default settings (*Metsalu and Vilo, 2015*). For venn diagrams a minimum of 2 'razer and unique' peptides in at least one replicate was required. Gene enrichment analysis was performed using the DAVID 6.7 functional annotation online tool (*Huang et al., 2009*) with an infected cellular lysate sample as the background or uninfected exosome protein list. Results were downloaded and plotted using the GOPlot (v.1.0.2, CRAN) (*Walter et al., 2015*) and ggplot2 (v.3.3.0, CRAN) libraries in R with Benjamini corrected p-values.

### Cryogenic transmission electron microscopy

Vesicle pellets from gradient fractions were resuspended in 30 μl HEPES buffered saline (0.85% (w/v) NaCl, 10 mM HEPES-NaOH, pH 7.4) on ice. Holey carbon 2/2 C-flat EM grid (Protochips) grids were glow-discharged for 30 s in $N_2$ plasma, and 2.5 μl of sample was applied to the surface at 4°C and 100% humidity in a FEI Vitrobot. Grids were blotted for 4 s using Whatman 595 blotting paper, then plunge-frozen in liquid ethane. Electron micrographs were acquired using a FEI Titan Krios TEM, operated at 300 KeV in EFTEM mode at a nominal magnification of 6400X and 81000X. Images were digitized using Gatan K2-Bioquantum GIF operated in 'Counting Mode'. Sample acquisition was performed in 'Low Dose' with area exposed to 5-50e⁻/A². Dimensions were measured using ImageJ.

### siRNA knockdown and RT-qPCR

MRC5 cells ($5 \times 10^4$) were seeded in 24-well plates 24 hr prior to transfection. 20 pmol of siRNA (siGENOME SMARTpool, Dharmacon) was diluted in Opti-MEM (Life Technologies), and transfected using Lipofectamine RNAiMAX (Thermo), according to the manufacturer's instructions. Cells were transfected twice, 24 hr apart, and then infected with HCMV 24 hr later. 5 DPI, cells were washed

with PBS, lysed with TRIzol Reagent (Invitrogen), and total RNA isolated according to manufacturer's instructions. DNase treatment of RNA and cDNA synthesis was performed using the RQ1 (Promega) and SuperScript III kit (Thermo), respectively, according to manufacturer's instructions. two step qPCR of all samples was performed using QuantiNova SYBR Green PCR master mix (Qiagen) in a Rotor-Gene-Q real time PCR cycler (Qiagen). Initial heat activation was performed at 95°C for 5 min, then 40 cycles of 30 s denaturation (95°C) and 60 s anneal/extend (60°C). GAPDH was used as the internal control, and relative transcript level quantified using the ddCt method. siRNA and qPCR primer sequences are listed in *Supplementary file 3*.

## IE1 fluorescent focus assay

5 DPI, infected cell supernatant was collected, cell debris removed by centrifugation at 500xg for 5 min, and volumes normalized to 500 μ. Serial dilutions using full media (1:4) were performed, starting from neat samples to $4^{-5}$, in technical duplicate, and 100 μl of each dilution added to a 96-well reporter plate of confluent uninfected MRC5 cells. 24 HPI, cells were fixed and stained for IE1 as per 'Infectivity assay of gradient fractions'. Reporter plates were automatically imaged using a DMi8 (Leica) microscope with 10x objective. A focus map was constructed with a single point per well, and autofocused using the Hoechst channel in LAS X navigator (Leica). A 3 × 3 tilescan was performed per well in hoechst, and IE1 channels with 0% image overlap and fill factor 75%. IE1 foci per well were manually counted in LAS X core offline version (Leica) at appropriate dilutions for IU/ml calculations.

## Stable cell line generation

A cDNA library was made as per 'siRNA knockdown and RT-qPCR' from wild type MRC5 cells. The VAMP3 gene was PCR amplified (Forward primer: 5'-TGACGAATTCATGTCTACAGGTCCA-3'; Reverse primer: 5'-TGAGGATCCTCATGAAGAGACAAC-3') with Phusion high fidelity polymerase (ThermoFisher Scientific), digested with EcoRI and BamHI, then cloned into pLXSN vector (Clontech). To generate MRC5 cells stably expressing VAMP3 or GFP-FLAG, insert containing plasmids were isolated (PureLink plasmid miniprep kit, Invitrogen), and Phoenix cells (purchased from ATCC, and mycoplasma negative by PCR) transfected twice 24 hr apart using Lipofectamine 3000 (Invitrogen), according to manufacturer's instructions. 24 hr post transfection, medium containing retroviral particles was added directly to MRC5 cells (two harvests, 24 hr apart). Cells were placed under selection (full media containing 400 μg/ml G-418, Gibco) for 6 days, changed at 48 hr intervals, and then passaged as per 'cells and virus'.

## Intracellular viral genome quantification

Cells were transfected and infected as per conditions in 'siRNA knockdown and RT-qPCR'. 5 DPI, cells were washed with PBS, lysed with TRIzol Reagent (Invitrogen), and total DNA isolated according to manufacturer's instructions. qPCR was performed as per 'siRNA knockdown and RT-qPCR' with UL83 specific primers (Forward: 5'-GTCAGCGTTCGTGTTTCCCA-3'; Reverse: 5'-GGGACACAA-CACCGTAAAGC-3') and ATG5 specific primers (Forward: 5'-TATTACCCTTTGATGCCTTTTTTTC-3'; Reverse: 5'-AACTTGTACCACCAATTCTAAAATG-3') for quantification and internal control respectively. The relative number of viral genomes was calculated using the ddCt method, normalized to the Opti-MEM control.

## Extracellular viral genome quantification

5 DPI, infected cell supernatants (103 μl) were thawed on ice, then DNase treated with 5 μl RQ1 DNase (Promega) for 90 min at 37°C. 5 μl STOP solution was added and samples incubated for 10 min at 65°C. Viral DNA was extracted using the Qiagen Virus Spin Kit, according to manufacturer's instructions. Genome copies in 5 μl eluate were quantified by qPCR using UL83 specific primers as for 'intracellular viral genome quantitation'. Ct values were normalized to a standard curve constructed from 1:10 serial dilutions of pLXSN-UL83 ($1.23 × 10^7 – 1.23 × 10^3$ copies) and genome copies/ml calculated (*Gault et al., 2001*).

## Immunofluorescence and confocal microscopy

MRC5 cells were grown on 12 mm glass coverslips (no. 1.5, Menzel) in 24-well plates. At the indicated timepoints cells were fixed, permeabilized and blocked as for 'Infectivity assay of gradient fractions'. Mouse anti-HCMV UL99 (Clone 10B4 (21)), mouse anti-HCMV gB, (Abcam ab6499), rabbit anti-GM130 (Abcam ab52649), mouse anti-GM130 (BD Biosciences 610822) and rabbit anti-VAMP3 (Abcam ab200657) primary antibodies were diluted 1:500 in block solution, and coverslips incubated for 60 min at RT, washed three times in PBST, and corresponding anti-mouse or anti-rabbit Alexa Fluor 568 or 633 (Invitrogen) secondary antibodies (1:1000) added for 30 min at RT. Coverslips were washed, and mounted on slides with ProLong Gold containing DAPI (Molecular Probes). Samples were stored at 4°C and protected from light until imaging. Confocal images were acquired using an inverted Leica SP5 using 63x UV corrected oil immersion objective and sequential scan settings of individual channels (Fluor 633, Fluor 568, eGFP and DAPI) with a line average of 4 and frame average 1. Images were viewed and analyzed in LAS X core offline version (Leica).

## Electron microscopy sample preparation and imaging

Cells were cultured on 6 mm sapphire discs, and treated with siRNA prior to infection with HCMV AD169 for 5 days at MOI = 1. Cryopreservation was performed by high pressure freezing using a Wohlwend HPF Compact 02 using 0.05 mm carriers (*Villinger et al., 2014*). Hexadecane was used as filler, and no cryoprotecting agent was used (*Walther et al., 2013*). Freeze substitution and resin embedding was performed using an AFS1 (Leica) with the following conditions: 24 hr, at −90°C in 0.5% $OsO_4$ in acetone (analytical grade), after that the temperature was increased to −10°C at a rate of 4 deg/hour (20 hr), and kept at −10°C for an additional 5 hr. Then $OsO_4$ was removed by three 10 min acetone washes and the temperature was raised to 5°C at a rate 5 deg/hour (3 hr). Resin infiltration (Epon 812, EMS14120) was performed at room temperature in 1.5 mL Eppendorf vials in five steps of 1 hr each, with increasing concentration of resin: 25% then 50% and 75% followed by 2 × 100%. Blocks were polymerized for 72 hr at 60°C, after polymerization the resin blocks were cooled with liquid nitrogen vapor and the discs were mechanically detached. The embedded monolayer of cells was sectioned with a diamond knife to produce 50 nm slices, which were deposited on formvar coated TEM grids. Prior to imaging with the TEM, sections were coated with 5 nm of carbon and examined with a JEM 1400Plus (Jeol, Japan) operated at 80 keV.

## Acknowledgements

We thank the Monash Micro Imaging Platform, the Ramaciotti Centre for Cryo-Electron Microscopy, and the Monash Biomedical Proteomics Facility. We also thank Prof. Ileana Cristea for providing access to mass spectrometry instrumentation (DP1DA026192). We would like to thank Prof. Stephen Turner, Dr. Nathan Croft, and Dr. Chris Andoniou for critical reading of the manuscript. DLT was supported by an Australian Government Research Training Program (RTP) Stipend and RTP Fee-Offset Scholarship through Monash University. This work was supported by the National Health and Medical Research Council of Australia (# APP1100737 to RAM).

## Additional information

### Funding

| Funder | Grant reference number | Author |
|---|---|---|
| National Health and Medical Research Council | APP1100737 | Rommel A Mathias |
| National Health and Medical Research Council | APP1037043 | Rommel A Mathias |

The funders had no role in study design, data collection and interpretation, or the decision to submit the work for publication.

## Author contributions
Rommel A Mathias, Conceptualization, Data curation, Formal analysis, Supervision, Funding acquisition, Investigation, Visualization, Methodology, Writing - original draft, Writing - review and editing; Declan L Turner, Data curation, Formal analysis, Validation, Investigation, Visualization, Methodology, Writing - original draft, Writing - review and editing; Denis V Korneev, Investigation, Methodology; John G Purdy, Formal analysis, Investigation; Alex de Marco, Supervision, Investigation, Methodology

## Author ORCIDs
Declan L Turner ⓘD https://orcid.org/0000-0002-7643-2796
Denis V Korneev ⓘD http://orcid.org/0000-0003-4350-8317
John G Purdy ⓘD https://orcid.org/0000-0001-7460-7567
Alex de Marco ⓘD http://orcid.org/0000-0001-6238-5653
Rommel A Mathias ⓘD https://orcid.org/0000-0003-3064-4897

## Decision letter and Author response
Decision letter https://doi.org/10.7554/eLife.58288.sa1
Author response https://doi.org/10.7554/eLife.58288.sa2

# Additional files

## Supplementary files
• Supplementary file 1. Glycerol-tartrate gradient purified virion proteomics – unshaved and proteinase k shaved.
• Supplementary file 2. OptiPrep gradient fraction proteomics – uninfected and HCMV infected.
• Supplementary file 3. siGENOME SMARTpool siRNA sequences and qPCR primer sequences.
• Transparent reporting form

## Data availability
Mass spectrometry RAW files have been deposited to the ProteomeXchange Consortium via the PRIDE partner repository with the identifier PXD015081.

The following dataset was generated:

| Author(s) | Year | Dataset title | Dataset URL | Database and Identifier |
|---|---|---|---|---|
| Turner DL, Korneev DV, Purdy JG, de Marco A, Mathias RA | 2020 | The host exosome biogenesis pathway is essential for human cytomegalovirus virion assembly and egress | https://www.ebi.ac.uk/pride/archive/projects/PXD015081 | PRIDE, PXD015081 |

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
