## [Decision Letter]

**Acceptance summary:**

The presence of host proteins within the Human Cytomegalovirus (HCMV) virion has been known for a long time, yet their functional relevance is largely unknown. This study performed an in-depth analysis of the composition of HCMV virions and shows that HCMV virions are enriched in exosome components. Further, evidence is provided that this association is specific, suggesting that exosome components play important roles for viral particle assembly and egress.

**Decision letter after peer review:**

Thank you for submitting your article "The host exosome pathway underpins biogenesis of the human cytomegalovirus virion" for consideration by *eLife*. Your article has been reviewed by three peer reviewers, and the evaluation has been overseen by a Reviewing Editor and Karla Kirkegaard as the Senior Editor. The following individual involved in review of your submission has agreed to reveal their identity: Qiana Matthews (Reviewer #3).

The reviewers have discussed the reviews with one another and the Reviewing Editor has drafted this decision to help you prepare a revised submission.

Summary:

The authors performed proteomic analysis of virions and exosomes from HCMV-infected cells. Virions and exosomes were then analyzed for both host and viral proteins, and particular emphasis was laid on the enrichment of exosome components in virions. Finally, the authors show that some of these cellular components of the virions are functionally important for virion morphogenesis (envelopment and release from cells). The existence of host proteins within the HCMV virion is not a new finding, but the functional relevance of these proteins was not yet defined, which is indeed a very complex question.

This work contributes to this under investigated area, but further controls would be needed to ensure that the conclusions drawn by the authors hold true. In addition, an in-depth discussion about the different classes of extracellular vesicles would need to be included to reach a more thorough scholarship.

Essential revisions:

All reviewers agreed that this is overall a very nice piece of work. However, they also all agreed that additional controls would be needed to ensure that the results are accurate. We think that it would be important to exclude that NIEPs and dense bodies, which are difficult to separate from virions, did not affect the results. It should be clearly shown that all of the constituents identified are present in the same structure. In other words, how sure are the authors that the cellular proteins they have identified are genuinely present in virions, as opposed to being present in vesicles that co-purify with virions?

In the Materials and methods section it is stated that dense bodies and NIEPs were separated from virions before an in-depth analysis of the virion composition. However, it is not clear whether contaminations of NIEPs/dense bodies may contribute to the abnormal exosomes that are described by the authors. In fact, there is no mention of the potential complicating role of dense bodies/virions in the main text of the manuscript. At the very least, this needs to be discussed, but preferably, further controls are included in the revised version, which would make this manuscript much more convincing.

Specific points that we would like you to address are the following:

1) EM images are provided of two of the density fractions (Figure 2E/F), yet a more thorough analysis is required to interpret the data from these fractions. EM needs to be performed on each fraction, and the presence of NIEPs, dense bodies, virions, and vesicles in each needs to be at least semi-quantitated. I.e. the authors only mention 'virions' in their analysis yet NIEPs and dense bodies will be present too, and have different densities and therefore may run in different fractions. It is also important to know what is NOT present in each fraction. I.e. fraction 7 contains virions, but does it also contain vesicles? The text says that is does (Results), yet the comments on Figure 2E imply that it doesn't? This is especially important given that fractions are combined for later analysis.

2) As pointed out above, an important element is the dissection of exosome and virion protein content. It is unclear whether various extracellular vesicles co-purified with virions in the glycerol-tartrate gradients that were used for the initial virion analyses. This is really important information if the data in this study is to be compared with others, which draw conclusions based on glycerol tartrate gradient purification. I.e. does the field need to be careful when drawing conclusions from glycerol-tartrate gradients due to contaminating vesicles?

3) The authors consistently refer to anything in fraction 7 (Figure 2) as being 'in the virion'. Yet in the text they state that this fraction contains larger/denser exosomes in addition to virions. Therefore, the text needs to be changed throughout to reflect the fact that they do not know whether these proteins are present in virions, or copurifying vesicles.

4) Figure 5. In order to conclude that it is specifically virion transport to the VAC that is affected by VAMP3 depletion, the authors would need to show that genome replication and packaging into the capsid are unaffected. Otherwise, this section needs to be rephrased.

5) Were biological replicates performed for all of the different proteomics experiments? Figure 3A implies that triplicate replicates were performed, but this information is not given for most experiments.

---

## [Author Response]

Essential revisions:All reviewers agreed that this is overall a very nice piece of work. However, they also all agreed that additional controls would be needed to ensure that the results are accurate. We think that it would be important to exclude that NIEPs and dense bodies, which are difficult to separate from virions, did not affect the results.

In the first set of experiments in the paper, virions, NIEPS, and dense bodies were separated using classical glycerol-tartrate gradients. Although we only focussed on analysis of virions in the initial submission, the revised version of the manuscript includes images of the gradients showing clear isolation of all 3 species (Figure 1—figure supplement 1A). Individual bands were extracted, and secondary gradients run to further enrich each species. Therefore, we are confident our virion preps had minimal contamination from NIEPs or DBs.

In addition, although not described/included in our initial submission, we optimized the input material from infected cells that was put onto the glycerol-tartrate gradients. Specifically, we compared input material that was: a) conditioned medium only, versus b) conditioned medium pooled with supernatant from sonicated infected cells (a commonly used method to increase yield). We report that conditioned medium only, contained significantly less DBs (Figure 1—figure supplement 1B-D), and therefore used this method/sample for mass spectrometry analysis of fractions from both glycerol-tartrate and Optiprep gradients. These additional results are also included in the revised manuscript.

It should be clearly shown that all of the constituents identified are present in the same structure. In other words, how sure are the authors that the cellular proteins they have identified are genuinely present in virions, as opposed to being present in vesicles that co-purify with virions?

Given that the only way to show that all constituents are present in the same structure would be to do immuno-gold EM on the hundreds of proteins identified by mass spectrometry, this is unfortunately impossible. Nonetheless, previous papers have confirmed the presence of exosome markers in virions using this method (PMID: 17760879, 19888988), supporting our discovery that host exosome proteins are genuine virion constituents, and not non-specific contaminants.

Furthermore, we are confident that our virion preps are not contaminated by vesicles that co-purify, based on our experiments using Proteinase K. This set of experiments was specifically designed to “shave off” any proteins on the virion surface that may mediate non-specific binding/contamination. Shaved virions were then re-isolated to gradients to ensure their integrity, before mass spectrometry analysis demonstrated that host exosome proteins remained robustly identified, and significantly enriched (Figure 1J).

See specific points #1, #2 and #3 below for further discussion points addressing concerns regarding contamination and co-purification.

In the Materials and methods section it is stated that dense bodies and NIEPs were separated from virions before an in-depth analysis of the virion composition. However, it is not clear whether contaminations of NIEPs/dense bodies may contribute to the abnormal exosomes that are described by the authors.

Images of glycerol-tartrate gradients in the revised manuscript (Figure 1—figure supplement 1A) show that DBs are-denser than virions, so it is unlikely that DBs are the viral exosomes we report (that are less dense than virions). Additionally, we used conditioned medium only (containing minimal DBs, Figure 1—figure supplement 1B-D), as the input onto the Optiprep gradients. See specific point #1 below for further discussion of DBs resolving on Optiprep gradients.

NIEPs are also not likely to be the source of the viral exosomes we report for two main reasons. Firstly, we did not see any by EM in fraction 3 of the infected gradient (Figure 2F). Secondly, since the only difference between NIEPs and virions is the lack of viral DNA in NIEPs, an identical protein profile is expected. However, the main finding of the viral exosome analysis is that they are enriched with viral Fc-gamma receptor homologue IR11/gp34, and uncharacterised transmembrane proteins US14 and IRL12. By contrast, virions/NIEPs are instead abundant with MCP and UL83 (Figure 3C).

In fact, there is no mention of the potential complicating role of dense bodies/virions in the main text of the manuscript. At the very least, this needs to be discussed, but preferably, further controls are included in the revised version, which would make this manuscript much more convincing.

As suggested, we have taken the opportunity to include new images and text in the Results section that demonstrates that we did evaluate the contribution of NIEPs and DBs in our purification workflows, and considered them when interpreting our results. We trust that these inclusions now contribute to a much more convincing revised manuscript.

Specific points that we would like you to address are the following:1) EM images are provided of two of the density fractions (Figure 2E/F), yet a more thorough analysis is required to interpret the data from these fractions. EM needs to be performed on each fraction, and the presence of NIEPs, dense bodies, virions, and vesicles in each needs to be at least semi-quantitated. I.e. the authors only mention 'virions' in their analysis yet NIEPs and dense bodies will be present too, and have different densities and therefore may run in different fractions. It is also important to know what is NOT present in each fraction. I.e. fraction 7 contains virions, but does it also contain vesicles? The text says that is does (Results), yet the comments on Figure 2E imply that it doesn't? This is especially important given that fractions are combined for later analysis.

We specifically selected glycerol-tartrate gradients for analysis because we knew they could effectively resolve the different densities of virions, NIEPs, and DBs (Figure 1—figure supplement 1A). This first set of experiments demonstrated that DBs are denser than virions, and that fractionating conditioned medium only (and not material released from sonicated cells) significantly reduced the number of DBs present in the sample (Figure 1—figure supplement 1B-D). Thus, we applied these principles to the analyses using Optiprep gradients. Similarly, they should also contain trace amounts of DBs (relative to virions) and that if any were present, they would be at the bottom of the gradient. To this end, analysis of infected fraction 10 on the OptiPrep gradients revealed a small amount of viral protein, but the fraction had zero infectivity (Figure 2D, Figure 2—figure supplement 1C and F). Comparison of proteins in this faction with the original DB proteome published by Varnum et al. (PMID: 15452216), showed a considerable overlap (Figure 2—figure supplement 4D). Therefore, we are confident that infected fraction 7 containing virions is not contaminated with DBs, and we have added this new analysis of DBs to the revised manuscript.

The glycerol-tartrate gradients revealed that NIEPs are slightly less dense than virions (Figure 1—figure supplement 1A), and this is expected because they do not contain viral DNA. Importantly, this is the only difference between NIEPs and virions, and their protein profile (capsid, tegument, and envelope) is the same. On this basis, if any NIEPs are present on the Optiprep gradient, this does not confound our analysis, and is of no consequence to our conclusions made regarding virions.

With respect to performing EM on every single fraction, we actually conducted EM analysis on more fractions in the initial submission. However, we were unable to observe sufficient numbers to perform even semi-quantitative comparisons. Therefore, we included only representative images in Figure 2E and F. We did, however, take the opportunity to quantify the relative numbers of virions, NIEPs and DBs in unfractionated conditioned medium only by cryo-EM. In agreement with the glycerol-tartrate experiments, few DBs were observed compared to virions and NIEPs (Figure 1—figure supplement 1C-D).

For a discussion of the constituents in fraction 7, please see specific point #3 below.

2) As pointed out above, an important element is the dissection of exosome and virion protein content. It is unclear whether various extracellular vesicles co-purified with virions in the glycerol-tartrate gradients that were used for the initial virion analyses. This is really important information if the data in this study is to be compared with others, which draw conclusions based on glycerol tartrate gradient purification. I.e. does the field need to be careful when drawing conclusions from glycerol-tartrate gradients due to contaminating vesicles?

Images of glycerol-tartrate gradients in the revised manuscript show that virions, NIEPs, and DBs are clearly resolved (Figure 1—figure supplement 1A). Additionally, the top-most band on the gradient represents cell-derived vesicles (Figure 1—figure supplement 1A). Therefore, we conclude that glycerol-tartrate gradients are not contaminated, and can sufficiently resolve cell-derived vesicles and virions.

To further validate this finding, we performed Proteinase K experiments to “shave off” any proteins on the virion surface and limit any non-specific binding/contamination. Most importantly, following shaving and re-isolation on another gradient, host exosome proteins remained robustly identified in virions.

Therefore, our results do not suggest that virions purified on glycerol-tartrate gradients are contaminated with other vesicles. Quite the contrary; exosome proteins are genuine virion constituents, and this was observed using glycerol-tartrate gradients and Optiprep gradients for virion isolation.

3) The authors consistently refer to anything in fraction 7 (Figure 2) as being 'in the virion'. Yet in the text they state that this fraction contains larger/denser exosomes in addition to virions. Therefore, the text needs to be changed throughout to reflect the fact that they do not know whether these proteins are present in virions, or copurifying vesicles.

To clarify, we identified the expression of some exosome markers in fraction 7 of the uninfected gradient. Given the measured density of the fraction (Figure 2—figure supplement 1A), we reasoned that they could be present in larger microvesicles. Technically, we did not report the presence of MVs in the infected gradient, this has been inferred by the reviewer, and it may or may not be valid.

Nonetheless, we concluded that fraction 7 of the infected gradient contained virions on the basis that it contained the most and the highest expression known virion proteins (Figure 2C), the highest infectivity of all fractions (Figure 2D), and verified virions by EM analysis (Figure 2E). It is clear that virions are dominant in this fraction, and that they would be significantly enriched compared to any other population that may co-isolate. If not, the magnitude of the three features just mentioned would be compromised.

In addition, and as stated in the text, we deliberately first performed comparison A (Figure 2G) to remove any MVs proteins from the exosome population. This allows us to make a direct comparison between uninfected exosomes and virions (Comparison D, Figure 2G), without the contribution from potential co-isolating MVs. We have added additional panels in Figure 2—figure supplement 4 in the revised manuscript to illustrate this refinement procedure more clearly.

4) Figure 5. In order to conclude that it is specifically virion transport to the VAC that is affected by VAMP3 depletion, the authors would need to show that genome replication and packaging into the capsid are unaffected. Otherwise, this section needs to be rephrased.

As requested, we performed additional experiments to assess viral genome replication. Compared to NTC cells, we observed 1.7-fold less genomes in infected cells lacking VAMP3 (Figure 5—figure supplement 1B). This negligible reduction in viral genome copies does not reconcile the 55-fold and 5200-fold reduction in infectious virus titre observed for AD169 and RCMV1158, respectively.

It is widely understood that viral genome replication is an essential prerequisite required for true late (gamma 2) gene expression (Mocarski, Fields virology 2013, Human herpesviruses: biology, therapy, and immunoprophylaxis 2007). Figure 5E (confocal) and Figure 5—figure supplement 1C (western blot) displays robust detection of UL99 in VAMP3 KD conditions with similar levels to NTC cells, indicating that viral DNA synthesis has occurred, and late genes expressed.

We have also included new EM imaging results in the revised manuscript. Figure 5I now shows infected cells with the nucleus juxtaposed to the vAC in both NTC and VAMP3 KD conditions. Whilst both cells show assembled capsids in the nucleus, only NTC cells have enveloped capsids (maturing virions) in the cytoplasm. In addition, we also performed new western blotting experiments that confirm MCP expression is similar in both NTC and VAMP3 KD cells (Figure 5—figure supplement 1C).

Finally, on the basis that VAMP3 is a cytoplasmic vesicle-associated SNARE protein, we reason that its absence prevents capsids entering the vAC for further maturation.

5) Were biological replicates performed for all of the different proteomics experiments? Figure 3A implies that triplicate replicates were performed, but this information is not given for most experiments.

Single replicates were performed for both the unshaved and proteinase K shaved virions extracted from glycerol-tartrate gradients. The infected cellular lysate proteomic analysis was performed in biological triplicate. OptiPrep gradient fractionation and MS analysis was performed in biological triplicate. This information has been added to the general text, Materials and methods section, and figure legends of the revised manuscript.